# Crosstalk between nitric oxide and retinoic acid pathways is essential for amphioxus pharynx development

Filomena Caccavale[1], Giovanni Annona[1], Lucie Subirana[2], Hector Escriva[2], Stephanie Bertrand[2], Salvatore D'Aniello[1]*

[1]Department of Biology and Evolution of Marine Organisms (BEOM), Stazione Zoologica Anton Dohrn Napoli, Napoli, Italy; [2]Sorbonne Université CNRS, Biologie Intégrative des Organismes Marins (BIOM), Observatoire Océanologique, Banyuls-sur-Mer, France

**Abstract** During animal ontogenesis, body axis patterning is finely regulated by complex interactions among several signaling pathways. Nitric oxide (NO) and retinoic acid (RA) are potent morphogens that play a pivotal role in vertebrate development. Their involvement in axial patterning of the head and pharynx shows conserved features in the chordate phylum. Indeed, in the cephalochordate amphioxus, NO and RA are crucial for the correct development of pharyngeal structures. Here, we demonstrate the functional cooperation between NO and RA that occurs during amphioxus embryogenesis. During neurulation, NO modulates RA production through the transcriptional regulation of *Aldh1a.2* that irreversibly converts retinaldehyde into RA. On the other hand, RA directly or indirectly regulates the transcription of *Nos* genes. This reciprocal regulation of NO and RA pathways is essential for the normal pharyngeal development in amphioxus and it could be conserved in vertebrates.

*For correspondence:
salvatore.daniello@szn.it

Competing interests: The authors declare that no competing interests exist.

## Introduction

The ontogenesis of the vertebrate head is a complex developmental process in which both neural crest and non-neural crest cells participate. The craniofacial development and the correct antero-posterior patterning of head structures are driven by complex interactions among several signaling pathways and epigenetic mechanisms (*Haworth et al., 2007*; *Jacox et al., 2014*; *Kong et al., 2014*; *Francis-West and Crespo-Enriquez, 2016*). In this context nitric oxide (NO) is a potent morphogen playing crucial roles in head structures development. Loss-of-function of neuronal nitric oxide synthase (*Nos1*) in *Xenopus* and zebrafish induces mouth opening arrest, smaller eyes, and substantial aberrations in cartilage and bone formation (*Jacox et al., 2014*). Moreover, inhibition of NO production is responsible for severe defects in pharyngeal arch patterning, consistent with the observation of alterations in the Hox code (*Kong et al., 2014*).

The development, as well as the antero-posterior and dorso-ventral patterning, of the head and pharynx shows conserved features within the chordate phylum. In amphioxus, which belongs to the cephalochordate subphylum, the pharynx is characterized by a marked left-right asymmetry which is controlled by the Nodal signaling pathway, namely by the Cerberus-Nodal-Lefty-Pitx cascade (*Bertrand et al., 2015*; *Soukup et al., 2015*; *Li et al., 2017*). The antero-posterior patterning and development of amphioxus pharyngeal slits are driven by a conserved set of transcription factor genes, among which *Hox1*, *Pax2/5/8*, *Pitx*, and *Tbx1/10*, that are also involved in vertebrate pharyngeal arches formation (*Schubert et al., 2005*; *Bertrand et al., 2015*; *Wang et al., 2019*).

In amphioxus, NO is enzymatically produced by synthases encoded by three distinct genes – *NosA*, *NosB*, and *NosC* – that are derived from cephalochordate-specific gene duplications and

show a complementary expression pattern during development (*Annona et al., 2017*; *Marlétaz et al., 2018*). During amphioxus embryogenesis, as a consequence of pharmacological inhibition of endogenous NO production, the opening of the mouth is prevented as well as the correct development of other important pharyngeal structures, such as the endostyle and the club-shaped gland (*Annona et al., 2017*). Moreover, the treated larvae show a posteriorized phenotype, resembling the well-described phenotype induced by exogenous retinoic acid (RA) administration during amphioxus embryogenesis (*Escriva et al., 2002*; *Schubert et al., 2005*; *Koop et al., 2014*). This experimental evidence prompted us to analyze the molecular effects of NO synthesis inhibition in detail by using a differential transcriptomic approach in the cephalochordate *Branchiostoma lanceolatum*. Our results show that the pharyngeal phenotype observed after reduction of endogenous NO production derives from an alteration of the RA signaling pathway. Moreover, we highlight the existence of a crosstalk between these pathways and propose that it is crucial to fine-tune the NO/RA balance that is required for proper pharyngeal development.

## Results

### NO controls pharyngeal development during early neurulation in amphioxus

Previous studies, using pharmacological inhibition approaches, have highlighted the involvement of NO in the specification of amphioxus pharyngeal structures during neurulation (*Annona et al., 2017*). To better characterize the key role of NO during embryonic development, we narrowed down the time window of pharmacological treatment by defining the exact timing during which an inhibition of endogenous NO production affects the development of the pharynx. Therefore, we performed short-term in vivo treatments with the Nos activity inhibitor 1-[2-(trifluoromethyl)phenyl]−1*H*-imidazole (TRIM) during *B. lanceolatum* development testing different drug exposure times between the early neurula stage (N2 stage, 24 hr post fertilization [hpf] at 18°C) and the pre-mouth larva stage (transition stage T1, 48 hpf at 18°C) (*Figure 1A*). During the selected time window, the endogenous NO mainly derives from the activity of NosC, whose gene expression is observed from the N2 stage until the larval stage (L1, 72 hpf at 18°C) (*Figure 1—figure supplement 1*; *Annona et al., 2017*). We cannot exclude a contribution to the NO production by NosB, whose gene is expressed from gastrula (G3 stage, 10 hpf at 18°C) to neurula (N2) (*Annona et al., 2017*), although we think it should be minimal because its mRNA synthesis runs out at 24 hpf. On the other hand, *NosA* is expressed in the adult and not during embryonic development.

The phenotype resulting from the different treatments was scored at the open-mouth L1 stage. The morphological alterations included: (i) a mean reduction of 31% of the pharynx length being overall unchanged the body length (*Figure 1B* and *Figure 1—figure supplement 2A,B*), (ii) the complete or partial absence of mouth opening on the left side of the pharynx (*Figure 1B*, panels I, III and V, VII), and (iii) the incomplete formation of the club-shaped gland and of the endostyle (*Figure 1B*, panels I, III and V, VII), the latter being positioned more ventrally than in controls (*Figure 1B*, panels II, IV and VI, VIII).

The inhibition of NO production during the 24–30 hpf time window at 18°C was the shortest treatment inducing a significant effect by producing 100% of abnormal larvae. Delayed treatments starting at 30 or 36 hpf, for 6, 12, or 18 hr, resulted in approximately 70% of affected larvae (*Figure 1A*). Moreover, the endogenous NO concentration was measured at the endpoints of TRIM incubation intervals (24–30, 24–36, and 24–42 hpf) and in comparison to controls it decreased by 66%, 57%, and 55%, respectively, demonstrating the efficiency of the treatments (*Figure 1—figure supplement 2C*). Conversely, when TRIM treatment was performed later, between 42 and 48 hpf, the larvae were not affected (*Figure 1A*). These results suggest that pharyngeal development is, at least in part, under the control of NO during neurulation (24–42 hpf) and we observed that the minimal time window of TRIM treatment that induces pharynx malformations corresponds to the first 6 hr (24–30 hpf).

The results presented here are slightly different from those previously published by our group (*Annona et al., 2017*). In our preceding study, we suggested that the developmental time window of Nos action was between 36 and 48 hpf because the treatment between 24 and 36 hpf was inefficient in inducing a phenotype. However, in the present work we show that a treatment between 24

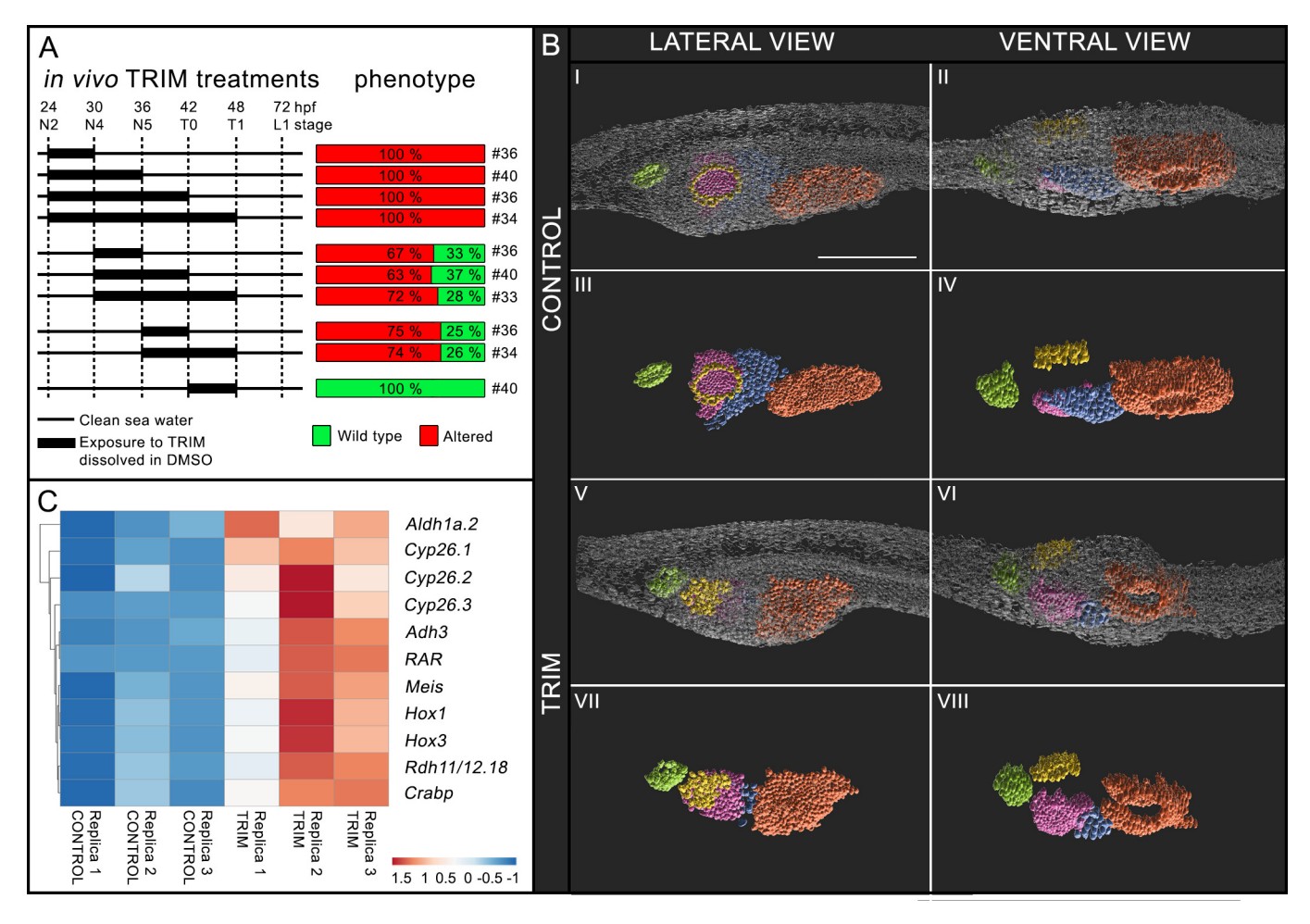

**Figure 1.** Characterization of in vivo 1-[2-(trifluoromethyl)phenyl]—1*H*-imidazole (TRIM) treatment during early amphioxus embryogenesis. (**A**) Schematic representation of time intervals during which embryos were grown in presence of TRIM and the resulting phenotype. (**B**) 3D reconstruction of control and TRIM-treated larvae showing anatomical alterations in pharyngeal region (panels I, II, V, VI). Only internal anatomical structures are highlighted in panels III, IV, VII, VIII. Larvae orientation: anterior to the left, dorsal to the top. Scale bar: 50 μm. Color code: green = pre-oral pit, violet = endostyle, yellow = mouth, blue = club-shaped gland, orange = gill slit. (**C**) Gene expression heatmap, for selected genes, of the differential transcriptomic analysis (control versus TRIM).

The online version of this article includes the following source data and figure supplement(s) for figure 1:

**Source data 1.** DESeq2 output for TRIM treated versus control condition.
**Figure supplement 1.** *NosC* expression pattern by whole-mount in situ hybridization.
**Figure supplement 2.** Pharynx and body lenght measurement and Nitric oxide quantification.
**Figure supplement 3.** RNA-seq data quality.
**Figure supplement 4.** Phylogenetic analysis for two retinoic acid (RA) pathway genes.
**Figure supplement 5.** Validation of the RNA-seq data by gene expression analyses.

and 30 hpf is fully penetrant. In *Annona et al., 2017*, the inhibition of NO synthesis was achieved by using the L-NAME (*N*ᵂ-nitro-L-arginine methyl ester), which is an L-arginine analog and is able to slow down the Nos activity. On the other hand, here we used TRIM, a molecule that is able to interfere with the binding of Nos enzymes with its substrate L-arginine and co-factor tetrahydrobiopterin. Thus, the dissimilarity in the time window of inhibition between the two treatments can be explained by the significant difference in Nos inhibition efficiency between the two molecules. Indeed, TRIM is effective at much lower concentrations than L-NAME (*Annona et al., 2017*), which probably justifies why a shorter treatment duration leads to the pharyngeal defect phenotype when we used TRIM, and hence the apparent discrepancy with previous data.

Based on the experimental evidence obtained in the present work, we performed a differential transcriptomic analysis comparing TRIM-treated N4 embryos (continuous treatment from 24 to 30 hpf) with wild-type embryos in order to define the genes acting downstream of NO signaling during pharyngeal development in amphioxus (*Figure 1C*, and *Figure 1—figure supplement 3*).

## Inhibition of NO synthesis in vivo induces up-regulation and ectopic expression of RA pathway genes

The differential RNA-seq analysis revealed the up-regulation of 392 genes and the down-regulation of 50 genes upon TRIM treatment (*Figure 1C*, and *Figure 1—figure supplement 3*). Interestingly, several differentially up-regulated genes are implicated in RA metabolism and signaling pathways (synthesis and storage, catabolism and known direct RA target genes): *Adh3, Rdh11/12.18, Aldh1a.2, Crabp, Cyp26.1, Cyp26.2, Cyp26.3, RAR, Hox1, Hox3, Meis* (*Figure 1C* and *Figure 1—figure supplement 4*). In order to confirm this finding, we additionally validated RNA-seq data by quantitative RT-PCR (qRT-PCR) analyses of up-regulated, down-regulated, and unaffected genes. The results showed a consistent expression trend with the RNA-seq data (*Figure 1—figure supplement 5A–C'*). Moreover, the expression pattern of RA target genes *Hox1, Hox3, Meis,* and that of *Cyp26* genes was further investigated by whole-mount in situ hybridization in both control and TRIM-treated embryos at the neurula N5 (36 hpf at 18°C) and pre-mouth T1 (48 hpf at 18°C) developmental stages. Such analyses showed that endogenous NO reduction produced an effect not only on the expression level of RA metabolism and signaling pathway genes, but also on the expression territories of most of them. The *Hox1, Hox3,* and *Meis* anterior limit of expression was shifted anteriorly in TRIM-treated embryos in comparison to controls, indicative of the embryo's body posteriorization (*Figure 2A*). The RA catabolism enzyme genes that are duplicated in amphioxus, *Cyp26.1, Cyp26.2,* and *Cyp26.3,* showed a heterogeneous behavior: *Cyp26.2* was slightly up-regulated and its expression pattern did not change after TRIM treatment, whereas *Cyp26.1* and *Cyp26.3* were strongly up-regulated and showed an ectopic expression (*Figure 2A,D*). In particular, after inhibition of NO production, *Cyp26.1* expression was shifted anteriorly, while *Cyp26.3* expression was shifted posteriorly. Moreover, *Cyp26.3* showed an additional domain of expression in the tailbud, mainly in T1 stage embryos (*Figure 2A*). As a negative control we tested the expression pattern of *Pitx, Six1/2, IrxC,* and *Cdx,* whose expression levels were not affected after TRIM treatment in the RNA-seq data. Likewise, we observed no modification of their expression pattern after treatment (*Figure 1—figure supplement 5D*).

### *Aldh1a.2* expression is specifically regulated by NO

The above-mentioned gene expression results (*Figure 2A,C,D*) suggested that the abnormal pharyngeal development of TRIM-treated embryos could be the result of an overactivation of the RA signaling pathway based on previous studies (*Escriva et al., 2002*; *Schubert et al., 2006*; *Koop et al., 2014*; *Carvalho et al., 2017b*). In order to test this hypothesis, we performed two in vivo experiments in parallel in which neurula stage embryos were incubated for 6 hr, from 24 to 30 hpf, in the presence of either TRIM or RA. Then, we analyzed the relative expression of three groups of genes by qRT-PCR, which we previously found to be up-regulated in the RNA-seq analysis: (i) genes involved in the synthesis and storage of RA (*Adh3, Rdh11/12.18, Aldh1a.2,* and *Crabp*); (ii) genes that mediate RA effects (*RAR, Hox1, Hox3,* and *Meis*); and (iii) genes involved in RA degradation (*Cyp26.1, Cyp26.2,* and *Cyp26.3*). All analyzed genes were up-regulated after both TRIM and RA treatment, with the exception of *Aldh1a.2* which was exclusively up-regulated after TRIM treatment (*Figure 2B,C,D*).

### *NosA* and *NosB* respond to exogenous RA during development

The expression analysis of the three amphioxus *Nos* genes after TRIM treatment revealed transcriptional up-regulation for two of them, *NosA* and *NosB,* while *NosC,* remained insensitive to the pharmacological treatment (*Figure 2E*).

In order to check if *NosA* and *NosB* up-regulation could be due to an indirect effect of the intracellular increase of RA caused by the TRIM treatment, we tested their expression levels after the addition of exogenous RA (*Figure 2E*). Similarly to the TRIM treatment, we observed that *NosA* and *NosB* expression were significantly up-regulated as a consequence of RA administration (*Figure 2E*).

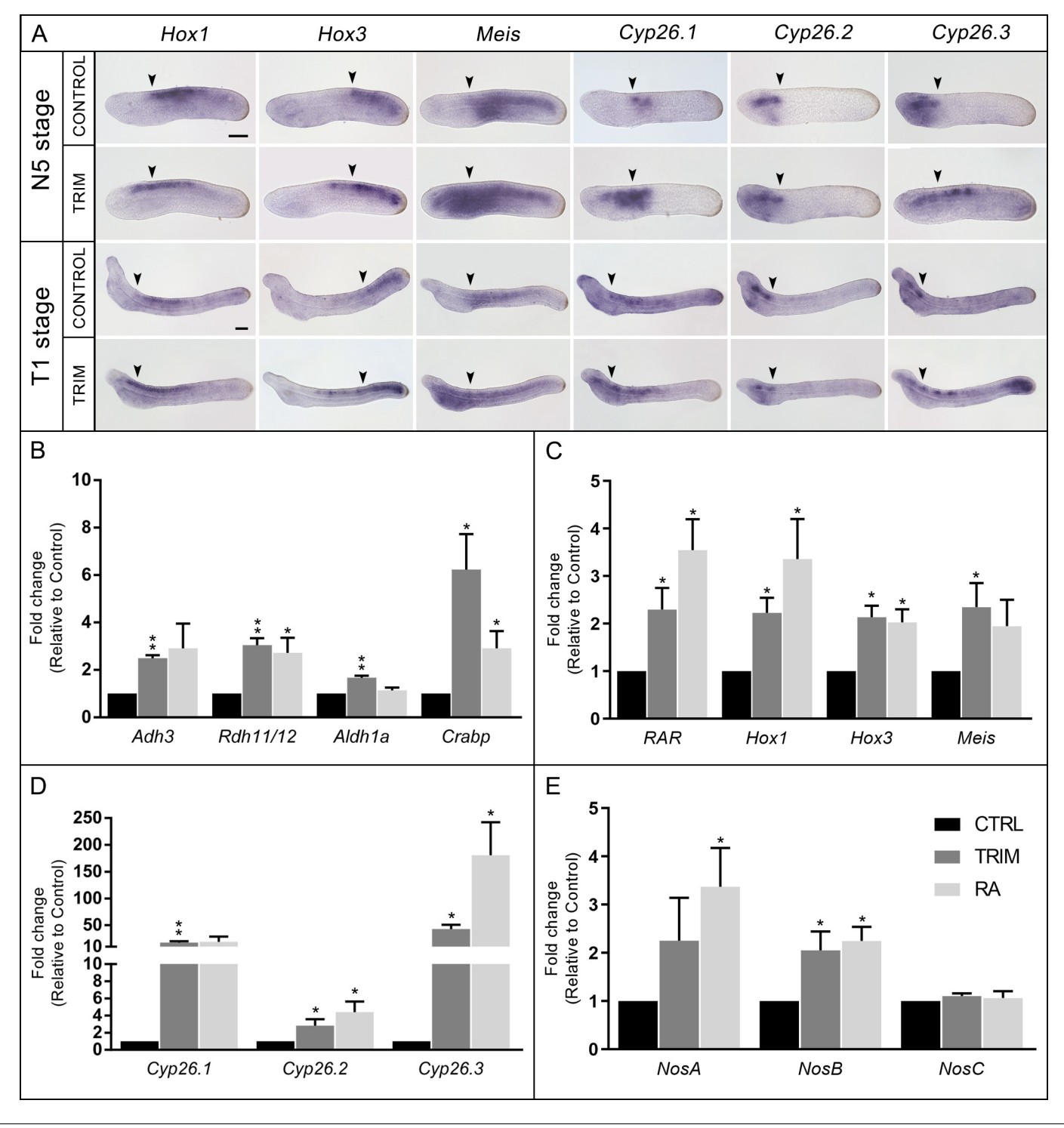

**Figure 2.** Analysis of gene expression. (A) Gene expression pattern by in situ hybridization of *Hox1*, *Hox3*, *Meis*, *Cyp26.1*, *Cyp26.2*, and *Cyp26.3* in 1-[2-(trifluoromethyl)phenyl]−1*H*-imidazole (TRIM)-treated and control embryos at N5 and T1 stage. The anterior (*Hox1*, *Hox3*, *Meis*, *Cyp26.1*) and posterior (*Cyp26.2* and *Cyp26.3*) limits of gene expression territories in wild-type embryos are indicated with arrowheads in both control and TRIM-treated conditions. Fifteen embryos were used for each probe and all showed the pattern presented here. Embryos orientation: anterior to the left, dorsal to the top. Scale bar: 50 μm. (B–E) Quantitative RT-PCR (qRT-PCR) on N4 stage embryos showing expression level changes after 6 hr of pharmacological TRIM or retinoic acid (RA) treatments (24–30 hpf) of: (B) genes encoding enzymes for RA synthesis: *Adh3*, *Rdh11/12.18*, *Aldh1a.2*, and binding protein for storage: *Crabp*; (C) RA direct target genes: *RAR, Hox1, Hox3, Meis*; (D) genes encoding RA degradation enzymes: *Cyp26.1, Cyp26.2, Cyp26.3*; (E) *Nos* genes: *NosA, NosB, NosC*. The statistical significance indicated is: * = p-value < 0.05; ** = p-value < 0.01.
*Figure 2 continued on next page*

*Figure 2 continued*

The online version of this article includes the following source data and figure supplement(s) for figure 2:

**Source data 1.** Source raw data of qRT-PCR in Figure 2B-E.

**Figure supplement 1.** *Nos* genes expression pattern by quantitative RT-PCR (qRT-PCR) after retinoic acid (RA) treatment.

Moreover, exogenous RA induced up-regulation of *NosA* expression up to 36 hpf (N5) and of *NosB* expression up to 48 hpf (T1) (*Figure 2—figure supplement 1*).

These results suggest a transcriptional control of RA on *Nos* genes expression during embryogenesis in amphioxus, but whether it is direct or indirect remains to be investigated.

## A RALDH inhibitor and a RAR antagonist are able to rescue the normal phenotype after inhibition of NO synthesis

To confirm that the up-regulation of *Aldh1a.2*, which could result in an endogenous RA increase, was the key event underlying pharyngeal alterations in TRIM-treated larvae, we performed two independent phenotypic rescue experiments using the retinaldehyde dehydrogenases (RALDH) inhibitor DEAB (*N,N*-diethylaminobenzaldehyde) and the RA antagonist BMS009. Both DEAB and BMS009 were applied in combination with TRIM to embryos at 24 hpf and removed at 30 hpf. As a control, the TRIM treatment was performed in parallel on another batch of embryos. The combined treatment with TRIM and DEAB resulted in a wild-type phenotype in 76% of the total observed larvae ('rescue' in *Figure 3A,B* panels I and II). On the other hand, 14% of the larvae showed an intermediate phenotype with normal pharynx length and organization in comparison to wild-type larvae, although the mouth was smaller and the club-shaped gland had an abnormal morphology ('partial rescue' in *Figure 3A,B* panels III and IV). The remaining larvae showed an affected phenotype as described above for the TRIM treatment ('altered' in *Figure 3A,B* panels V and VI). Similarly, the phenotype rescue experiment performed using the combination of TRIM and BMS009 led to 54% of wild-type larval morphology and 21% of larvae with a smaller mouth (*Figure 3A*). Moreover, the morphological rescue obtained by TRIM+DEAB treatment was associated with the rescued expression pattern of RA catabolism (*Cyp26.1* and *Cyp26.3*) and RA target genes (*Hox1*, *Hox3*, and *Meis*) (*Figure 3C*) at the N5 neurula stage. Therefore, by using two independent experiments, we demonstrated that the reduction of RA pathway activity in TRIM-treated embryos rescued the wild-type phenotype, suggesting that the observed effects of the inhibition of NO synthesis are produced by an increase in RA signaling.

## Discussion

### NO controls the RA concentration

The inhibition of NO production during amphioxus neurulation affects the normal formation and patterning of pharyngeal structures at the larva stage, including the length of the pharynx. From a molecular point of view, the differential RNA-seq approach revealed a clear up-regulation of different RA pathway players after Nos activity inhibition, suggesting that such de-regulation is responsible for the observed phenotype. The role of RA in pharyngeal morphogenesis has been extensively described in the literature; RA acts through *Hox1* in establishing the posterior limit of amphioxus pharynx. *Hox1* is co-expressed with the RA receptor (RAR) in the midgut endoderm and, in turn, represses the expression of pharyngeal endoderm markers, such as *Pax1/9* and *Otx* (*Schubert et al., 2005*). Nevertheless, the formation of pharyngeal slits requires low levels of RA. This condition is guaranteed: (i) by the activity of RA degradation enzymes (Cyp26), (ii) by the expression of TR2/4, a transcriptional repressor which binds to Retinoic Acid Response Elements (RARE) and decreases RA signaling in the anterior part of the animal, and (iii) by the fact that the central region of the embryo producing RA moves posteriorly as the embryo elongates (*Escriva et al., 2002*; *Koop et al., 2014*). In the present work, as a result of the inhibition of Nos activity, we observed the up-regulation of RA target genes, *Hox1*, *Hox3* and *Meis,* and that their anterior limit of expression shifted anteriorly. Furthermore, the RA degrading enzyme genes, *Cyp26.1* and *Cyp26.3*, were also sensitive to the inhibition of endogenous NO production showing an increased and ectopic expression. Cyp26 are required for RA degradation in the endoderm and ectoderm and have a key role in the

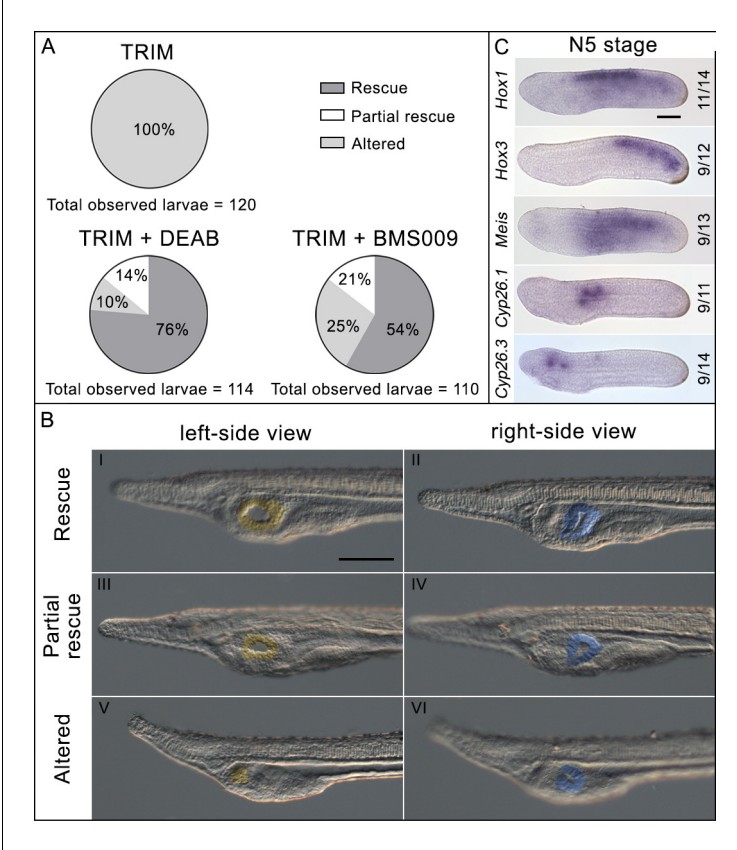

**Figure 3.** Phenotypic rescue effect of *N,N*-diethylaminobenzaldehyde (DEAB) and BMS009 on 1-[2-(trifluoromethyl) phenyl]−1*H*-imidazole (TRIM)-treated embryos. (**A**) Pie charts of the phenotypes observed after TRIM treatment and the combinatorial pharmacological treatments TRIM (100 µM) + DEAB (25 µM) or TRIM (100 µM) + BMS009 (10⁻⁶ M). The percentages of each observed phenotype are reported in the respective portions of the graphs. For each treatment, the total number of observed larvae is indicated below the chart. (**B**) Pictures of the pharyngeal region of larvae presenting the three different classes of phenotype observed in the rescue experiments: rescue, partial rescue, and altered. The mouth is highlighted in yellow and the club-shaped gland in blue. Larvae orientation: anterior to the left, dorsal to the top. Scale bar: 50 µm. (**C**) Expression pattern by in situ hybridization of *Hox1*, *Hox3*, *Meis*, *Cyp26.1*, and *Cyp26.3* after rescue assay with DEAB showing the restoration of wild-type expression territories. Numbers indicate the ratio between embryos showing a restored expression pattern and the total number of embryos analyzed. Embryo orientation: anterior to the left, dorsal to the top. Scale bar: 50 µm.

establishment and maintenance of the antero-posterior RA concentration gradient in amphioxus (*Carvalho et al., 2017a*). The up-regulation of *Cyp26* genes is a known consequence of RA excess, which is responsible for the posteriorization of larval body structures and for the pharynx loss (*Escriva et al., 2002*; *Schubert et al., 2005*; *Schubert et al., 2006*; *Minoux and Rijli, 2010*; *Koop et al., 2010*; *Koop et al., 2014*; *Bertrand et al., 2015*; *Carvalho et al., 2017b*). Altogether, these results suggest that the observed phenotype in TRIM-treated amphioxus embryos could be due to an increase in RA production. In previous studies, it has been shown that exogenous RA administration disrupts its endogenous gradient and causes the whole body posteriorization (*Escriva et al., 2002*; *Koop et al., 2014*; *Osborne et al., 2009*), highlighted for example by the anterior shift of *Cdx* expression. Here, instead, we show that the TRIM treatment only partially phenocopies exogenous RA application, resulting in a local posteriorization mainly restricted to the anterior part of the body including the pharynx area, while the posterior region is unaffected as demonstrated by the unaltered *Cdx* expression (*Figure 1—figure supplement 5D*). Therefore, this suggests the occurrence of an NO-mediated regulation of the RA pathway in anterior tissues versus an NO-independent mechanism in the posterior tissues.

Intracellular RA is synthesized by the reversible oxidation of retinol into retinaldehyde by either alcohol dehydrogenases (ADH) or retinol dehydrogenases (RDH). Subsequently, retinaldehyde is irreversibly oxidized to RA by RALDH, mainly by ALDH1A (*Gallego et al., 2006*; *Duester, 2008*). In our experiments we observed a transcriptional up-regulation of *Adh3*, *Rdh11/12.18*, and *Aldh1a.2* after TRIM treatment. While a unique ortholog of vertebrate *Adh* genes, *Adh3*, has been identified in amphioxus (*Cañestro et al., 2002*); 22 *Rdh11/12* genes, derived from a lineage-specific expansion, were identified. These *Rdh11/12* genes are related to human *Rdh11*, *Rdh12, Rdh13*, and *Rdh14*, that together with *Rdh10* correspond to retinaldehyde reductases predominantly involved in retinoid metabolism and homeostasis (*Albalat et al., 2011*; *D'Aniello et al., 2015*; *Figure 1—figure supplement 4B*). For *Aldh1*, a total of six genes were identified in amphioxus, orthologs of human *Aldh1A1-3,* which are major players in the oxidation of RA (*Cañestro et al., 2006*; *Figure 1—figure supplement 4A*). In our study, *Adh3* and *Rdh11/12.18* genes were also up-regulated after administration of exogenous RA, suggesting a feedback regulation of RA synthesis, at least on reversible enzymatic steps. On the other hand, *Aldh1a.2* was insensitive to exogenous RA administration. Based on these results, we hypothesize that, under physiological conditions, NO transcriptionally regulates *Aldh1a.2* and, as a consequence, controls the production of endogenous RA. Further evidence to support our hypothesis is provided by the two independent rescue experiments. Technically, in association with TRIM, we used two drugs that specifically act on the most crucial steps of RA signaling pathway: an inhibitor of the RALDH enzymes activity that would compensate the excess of Aldh1a.2 protein, and an RA antagonist, which is able to compensate the RA over-production through its binding to RAR. In both experiments we observed a wild-type phenotype at the larva stage as a consequence of the recovery of the normal development, further supported by the restoration of the proper expression pattern of both RA-target and RA-degrading enzyme genes.

Based on our results, we propose that NO plays a key role in the regulation of RA level during neurulation in amphioxus by fine-tuning the expression of *Aldh1a.2*, keeping RA concentration within the optimal range. This precise balance between intracellular concentrations of NO and RA guarantees the correct expression level and localization of all RA downstream target genes. When NO is removed from the system, the RA metabolism machinery malfunctions, resulting in a cascade of events leading to the up-regulation of the entire RA signaling pathway.

The missing piece of the puzzle, therefore, seems to be an unknown molecular link, which is able to explain the control of *Aldh1a.2* transcription by NO. In other chordates, it has been demonstrated that the mechanisms by which NO regulates transcription of target genes are: (i) the control of the extracellular-regulated kinase (ERK) and the Mitogen-activated protein kinase (MAPK) phosphatases activity and, as a result, the modulation of phosphorylation or dephosphorylation of target transcription factors (*Castellano et al., 2014*), and (ii) the direct modulation of target proteins, like transcription factors, histone acetyltransferases, and deacetylases or DNA methyltranserases, through *S*-nitrosylation of specific cysteine residues (*Bogdan, 2001*; *Nott et al., 2008*; *Sha and Marshall, 2012*).

Thus, NO could control *Aldh1a.2* expression by modulating the phosphorylation or *S*-nitrosylation of a specific transcription factor/chromatin remodeling protein, the nature of which still requires further research to be discovered.

It would be important to improve this knowledge since very little information, restricted to vertebrates, is reported on the control of RA metabolism by NO. Some cytochrome P450 enzymes, involved in RA metabolism, were identified as putative NO-regulated proteins, but no evidence about putative transcriptional regulation has been reported so far (*Lee et al., 2014*; *Lee et al., 2017*).

## Crosstalk between NO and RA signaling pathways

The exogenous administration of RA induces the expression of amphioxus *NosA* and *NosB* that are normally not expressed during the developmental time window investigated in this study (*Annona et al., 2017*). Moreover, such transcriptional regulation is maintained throughout the critical time period during which NO is necessary for pharyngeal development (i.e. 24–42 hpf, *Figure 2—figure supplement 1*). Conversely, *NosC* was not affected by the increase of RA level. A possible explanation could be that while in normal conditions RA does not regulate *NosA* and *NosB* expression, it directly or indirectly regulates such gene expression in the case of an NO/RA imbalance as a

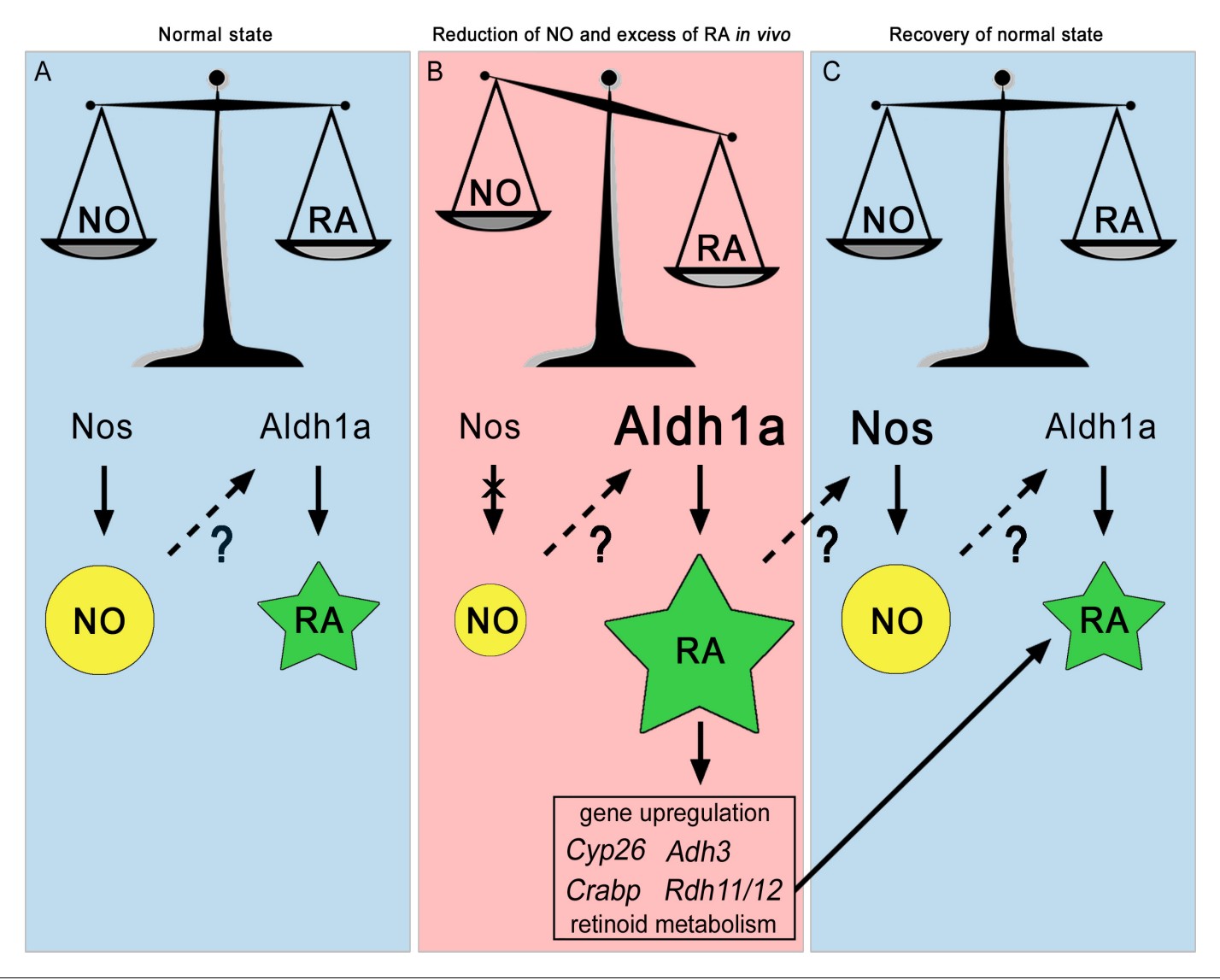

**Figure 4.** The NO:RA hypothesis. Schematic representation of the possible crosstalk occurring between Nitric Oxide (NO) and Retinoic Acid (RA) during chordate development. NO is represented by a yellow circle, RA by a green star, the ratio between NO and RA by a scale symbol. The decrease of NO and the increase of RA, Aldh1a and Nos, are indicated by different sizes of symbols relative to the normal state. Arrows represent enzymatic processes, while discontinuous arrows correspond to transcriptional regulation. Question marks indicate that it is not known exactly how NO regulates *Aldh1a.2* transcription and RA regulates *Nos* transcription. The cross over the arrow between Nos and NO illustrates the inactivation of Nos.

way to restore the correct NO/RA ratio. Additionally, it could be that RA is able to force the regulatory mechanism normally used at another developmental stage or in the adult.

In vertebrates, the role of NO and RA in the correct development of the pharynx and craniofacial structures has already been described (*Abe et al., 2008*; *Liu et al., 2013*; *Jacox et al., 2014*; *Chawla et al., 2018*). However, the existence of a possible crosstalk between NO and RA pathways has not yet been revealed. As mentioned above, there are few published data on the regulatory effect of NO on RA pathways, while the control of RA on NO production in vertebrates is more documented. For instance, the inhibitory or activator effect of RA on NO production through the activation of inducible and endothelial Nos, at both the protein and transcript levels, has been demonstrated in human cell lines (*Sirsjö et al., 2000*; *Achan et al., 2002*; *Hattori et al., 2002*; *Behairi et al., 2015*; *Moon, 2019*).

To summarize the regulatory loop between NO and RA signaling pathways in amphioxus, we formulated the NO:RA hypothesis, shown in *Figure 4*, arguing that their in vivo crosstalk is critical to maintain NO and RA concentrations in a correct ratio for normal pharyngeal development. During embryogenesis (*Figure 4A*), the balance of these two signaling molecules guarantees a physiological RA concentration necessary for a correct pharynx development. A reduction of NO (*Figure 4B*), as shown by our results albeit indirectly, induces an excess of RA, through the overexpression of *Aldh1a.2*. We propose that in the developing embryo an excess of RA compared to NO level induces both the expression of genes that contribute to RA level reduction, and to the overexpression of *Nos* genes resulting in NO production increase (*Figure 4C*) which in turn contributes to the decrease in RA concentration through an unknown mechanism. Thus, this regulatory loop would allow a reestablishment of the normal NO/RA ratio (*Figure 4C*) and would contribute to a correct development of the amphioxus pharynx.

## Conclusions

Our results show the existence of a functional crosstalk between NO and RA signals in the pharyngeal region of the cephalochordate amphioxus during neurulation. This opens new questions about the evolutionary conservation of this regulatory loop in vertebrates.

The role of RA, as well as that of NO, in amphioxus development and antero-posterior patterning of the pharynx has been described in several studies. Our results suggest the occurrence of a regulatory crosstalk between these two ancient and essential signaling pathways that has previously been neglected. Our results allow us to propose that, during amphioxus development, a precise NO/RA balance is necessary for the correct antero-posterior patterning of the pharynx. This endogenous intracellular balance is preserved by the reciprocal regulation of NO and RA pathways. Taking into account the role that these two signaling pathways and their crosstalk have for the correct development of the amphioxus pharynx, as well as the known role that, separately, has been demonstrated for these two pathways in vertebrates, we hypothesize that the functional and regulatory crosstalk between NO and RA pathways could be a conserved feature in vertebrates.

# Materials and methods

### Key resources table

| Reagent type (species) or resource | Designation | Source or reference | Identifiers | Additional information |
|---|---|---|---|---|
| Strain, strain background, (*Branchiostoma lanceolatum*) | Wild type | Collected in Argelès-sur-mer, France | NCBI Taxon: 7740 | |
| Chemical compound, drug | 2,3-Diaminonaphthalene (DAN) | Sigma-Aldrich | D2757 | |
| Chemical compound, drug | Nitrate reductase (NAD[P]H) from *Aspergillus niger* | Sigma-Aldrich | N7265 | |
| Chemical compound, drug | Flavin adenine dinucleotide disodium salt hydrate (FAD) | Sigma-Aldrich | F6625 | |
| Chemical compound, drug | β-Nicotinamide adenine dinucleotide 2′-phosphate reduced tetrasodium salt hydrate (NADPH) | Sigma-Aldrich | N7505 | |
| Chemical compound, drug | 1-[2-(Trifluoromethyl) phenyl]-1H-imidazole (TRIM) | Cayman chemical | 81310 | |
| Chemical compound, drug | *all-trans*-Retinoic acid (RA) | Sigma-Aldrich | R2625 | |
| Chemical compound, drug | N,N-diethylamino benzaldehyde (DEAB) | Sigma-Aldrich | D86256 | |

### Amphioxus embryos collection

Ripe adult European amphioxus (*B. lanceolatum*) were collected in Argelès-sur-mer (France) with a specific permission delivered by the Prefect des Pyrénées-Orientales. *B. lanceolatum* is not a

protected species. Spawning was induced during late spring and beginning of summer by employing a thermal shock as described by *Fuentes et al., 2007*. After in vitro fertilization, embryos were cultured in 0.22 μm filtered seawater at 18°C in plastic Petri dishes. According to the recent amphioxus ontology and staging (*Bertrand et al., 2021*; *Carvalho et al., 2021*) at 18°C, 24 hpf corresponds to neurula two stage (N2), 30 hpf to neurula four stage (N4), 36 hpf to neurula five stage (N5), 42 hpf to transition 0 stage (T0), 48 hpf to transition one stage (T1), and 72 hpf to larva 1 (L1). Embryos at desired developmental stages were incubated in the presence of specific pharmacological drugs, frozen in liquid nitrogen and kept at −80°C for subsequent RNA extraction or fixed with 4% paraformaldehyde in MOPS buffer overnight at 4°C and then stored in 70% ethanol at −20°C until use.

## Pharmacological treatments

Amphioxus embryos were treated at different developmental stages with the Nos inhibitor 1-[2-(trifluoromethyl)phenyl]−1H-imidazole (TRIM), with the RALDH inhibitor N,N-diethylaminobenzaldehyde (DEAB), with the RA antagonist BMS009 and with all-*trans* RA. All the drugs were dissolved in dimethyl sulfoxide (DMSO), and control embryo groups for each treatment were prepared adding an equal amount of DMSO. For TRIM treatments, a final concentration of 100 μM was used. For RA treatments, a final concentration of $10^{-6}$ M was used. All the treatments were performed in biological triplicates.

For rescue experiments, embryos at 24 hpf were treated simultaneously with a combination of 100 μM TRIM and 25 μM DEAB, or 100 μM TRIM and $10^{-6}$ M BMS009. At 30 hpf they were rinsed in filtered seawater and allowed to develop until the 72 hpf stage when the phenotype was observed. The experiment was performed in biological duplicates.

## Imaging

Control and TRIM-treated larvae at 72 hpf were stained with DAPI. A high-resolution Z-stack was acquired using a Zeiss confocal microscopy LSM 800, and a medium speed fast interactive deconvolution was applied. The 3D reconstruction was made employing Imaris 9.3.1 software; different larval body structures have been stained using the following five color blindness-friendly colors: #A6D854 green; #E78AC3 violet; #FFD92F yellow; #8DA0CB blue; #FC8D62 orange. Pharynx and body length measurements were performed on 72 hpf larvae captured with Axio Imager.Z2 microscope using a 10× objective, and employing the Ruler tool in Photoshop CS5 with a digital zoom of 50%. The pharynx was measured from the most anterior part of the pre-oral pit to the most posterior part of the first pharyngeal slit. An unpaired t-test was applied for the statistical analysis.

## Fluorimetric determination of endogenous NO concentration

Control and TRIM-treated frozen embryos at different developmental stages (from 24 hpf to 30, 36, and 42 hpf) were homogenized in PBS, sonicated (3 cycles of 1 min) and centrifuged at 20,000 *g* for 30 min at 4°C. The supernatants were collected for NO level analysis; 80 μl of each sample was incubated for 1 hr at room temperature in the presence of the nitrate reductase (0.06 U/ml), 2.5 μM FAD, and 100 μM NADPH. Then, 10 μl of 2,3-diaminonaphthalene (DAN) (0.05 mg/ml in 0.62 M HCl) were added and the samples were incubated for 15 min in the dark. The fluorescent product was stabilized in 1 N sodium hydroxide. Fluorescence was measured using the spectrofluorometer (Tecan) with excitation and emission at 365 and 425 nm, respectively, adding water up to a final volume of 200 μl. The results were normalized on the protein content. Total protein concentration was determined by the Bradford assay using a Bio-Rad Protein Assay Reagent (Bio-Rad) and bovine serum albumin as a standard.

## RNA-seq analysis

Total RNA was extracted from embryos using the RNeasy Plus Mini Kit (Qiagen) after sample homogenization using the TissueLyser (Qiagen). The RNA integrity number (RIN) was assessed by using TapeStatio4200 while RNA concentration and purity were estimated using a Nanodrop spectrophotometer. Indexed libraries were prepared from 1 μg/ea purified RNA with TruSeq Stranded Total RNA Library Prep Kit. Libraries were quantified using the Agilent 2100 Bioanalyzer (Agilent Technologies) and pooled so that each index-tagged sample was present in equimolar amounts, with a final concentration of 2 nM. The pooled samples at a final concentration of 10 pM were

subjected to cluster generation and sequencing using an Illumina NextSeq500 System in a 1×75 single read format (30 million reads). The raw sequence files generated (fastq files) underwent quality control analysis using FastQC. Transcriptome sequences were deposited in the NCBI Sequence Read Archive (SRA) database with the accession number: PRJNA630453.

Reads were mapped on the *B. lanceolatum* transcriptome (*Oulion et al., 2012*) using the aligner Bowtie2 with default parameters (*Langmead and Salzberg, 2012*). The read counts were obtained using IdxStats (*Li et al., 2009*; *Cock et al., 2013*) and the differential expression analysis between treated and wild-type embryos was performed using the R package DESeq2 (*Love et al., 2014*). Mapping and read counting were performed on the Roscoff ABiMS Galaxy platform.

## Phylogenetic analysis

Protein alignments were generated with ClustalX program using the sequence database reported in *Handberg-Thorsager et al., 2018*. Phylogenetic trees were reconstructed using maximum likelihood inferences calculated with PhyML v3.0 (*Guindon et al., 2010*).

## Gene expression analysis by whole-mount in situ hybridization and immunostaining

*Hox1*, *Hox3*, *Meis*, *Cyp26.1*, *Cyp26.2*, *Cyp26.3*, *Pitx*, *Six1/2*, *IrxC*, *Cdx*, and *NosC* were cloned in pGEM-T vector (Promega) using primers listed in *Supplementary file 1*. Antisense labeled riboprobes were synthesized and in situ hybridizations were performed as previously described (*Annona et al., 2017*; *Carvalho et al., 2021*). Whole-mount immunostaining of acetylated tubulin using monoclonal antibody produced in mouse (6-11B-1, Sigma) was performed as previously described in *Coppola et al., 2018*. Embryos were mounted in 80% glycerol in PBS and photographed using an Axio Imager.Z2 or a confocal microscope Zeiss LSM700.

## Gene expression analysis by qRT-PCR

Total RNA was extracted from embryos at different developmental stages: 30, 36, 42, 48, and 72 hpf using the RNeasy Plus Mini Kit (Qiagen); 350–1000 ng of total RNA were retrotranscribed in cDNA which was used undiluted (only for *Nos* genes) or diluted 1:10 for the qRT-PCR. Each reaction contained a final concentration of 0.7 µM of each primer and Fast SYBR Green Master mix with ROX (Applied Biosystems) in 10 µl total volume. qRT-PCR were run in a ViiA 7 Real-Time PCR System (Applied Biosystems). The cycling conditions were: 95℃ for 20 s, 40 cycles with 95℃ for 1 s, 60℃ for 20 s, 95℃ for 15 s, 60℃ 1 min, followed by a dissociation curve analysis using a gradient from 60℃ to 95℃ with a continuous detection at 0.015℃/s increment for 15 min. The results were analyzed using the ViiA 7 Software and exported into Microsoft Excel for further analysis. Each sample was processed in biological triplicates. The $2^{-\Delta\Delta Ct}$Rpl32 method was used to calculate the relative gene expression. Ribosomal protein L32 (*Rpl32*), expressed at a constant level during development, was used as a reference gene for the normalization of each gene expression level (*Annona et al., 2017*). Primers used are listed in *Supplementary file 1*. For the statistical analysis, we used the GraphPad Prism software employing the paired t-test. Statistical significance cut-off criteria was set at $p < 0.05$.

## Acknowledgements

The authors thank Rebecca Adikes and Hannah Rosenblatt, course assistants at the MBL Embryology Course 2019 in Woods Hole (USA), and Periklis Paganos for their help with imaging. We are also grateful to Enrico D'Aniello, Ricard Albalat, Marion Picard, and Haley Flom for their critical reading of the manuscript. We thank Ángel R de Lera for providing the RA antagonist BMS009, and Carola Murano and Anna Palumbo for their help with the NO quantification (DAN assay). We are grateful to the Institut Français de Bioinformatique and the Roscoff Bioinformatics platform ABiMS for providing computing and storage resources for the RNA-seq analysis, and the BIO2MAR platform (EMBRC-France) supported by ANR grant no. ANR-10-INBS-02 for giving us access to analytical material. SD and FC acknowledge the Assemble Plus project (contract numbers BA010618 and 360BA0619) and The Company of Biologists (grant number DEVTF-170211; sponsoring journal: *Development*) for supporting research visits to the Observatoire Océanologique of Banyuls-sur-Mer (France). FC was

supported by an OU-SZN PhD fellowship. SB is supported by the Institut Universitaire de France (IUF). HE is supported by the Centre national de la recherche scientifique and Agence Nationale de la Recherche (ANR) grants number ANR-16-CE12-0008-01 and ANR-19-CE13-0011. HE is supported by the European Commission ASSEMBLE Plus network (H2020-INFRAIA-1-2016–2017; grant number 730984).

## Additional information

### Funding

| Funder | Grant reference number | Author |
|---|---|---|
| European Commission | BA010618 and 360BA0619 | Salvatore D'Aniello |
| Company of Biologists | DEVTF-170211 | Filomena Caccavale |
| Agence Nationale de la Recherche | ANR-16-CE12-0008-01 and ANR-19-CE13-0011 | Hector Escriva |
| European Commission | H2020-INFRAIA-1-2016–2017 n. 730984 | Hector Escriva |

The funders had no role in study design, data collection and interpretation, or the decision to submit the work for publication.

### Author contributions

Filomena Caccavale, Conceptualization, Data curation, Investigation, Methodology, Writing - original draft, Writing - review and editing; Giovanni Annona, Data curation, Investigation, Methodology, Writing - review and editing; Lucie Subirana, Data curation, Formal analysis, Methodology; Hector Escriva, Conceptualization, Writing - review and editing; Stephanie Bertrand, Conceptualization, Data curation, Writing - review and editing; Salvatore D'Aniello, Conceptualization, Supervision, Investigation

### Author ORCIDs

Giovanni Annona http://orcid.org/0000-0001-7806-6761
Salvatore D'Aniello https://orcid.org/0000-0001-7294-1465

### Decision letter and Author response

Decision letter https://doi.org/10.7554/eLife.58295.sa1
Author response https://doi.org/10.7554/eLife.58295.sa2

## Additional files

### Supplementary files

• Supplementary file 1. Primers for the preparation of WISH probes cloning and for quantitative RT-PCR.

• Transparent reporting form

### Data availability

Sequencing data have been deposited in NCBI SRA under accession code PRJNA630453. All data generated or analysed in this study are included in the manuscript and supporting files. Source data files have been provided for Figures 1 and 2.

The following dataset was generated:

| Author(s) | Year | Dataset title | Dataset URL | Database and Identifier |
|---|---|---|---|---|
| Caccavale F, Annona G, Subirana L, Escriva | 2020 | Crosstalk between Nitric Oxide and Retinoic Acid pathways is essential for amphioxus pharynx | https://www.ncbi.nlm.nih.gov/sra/PRJNA630453 | NCBI Sequence Read Archive, PRJNA630453 |

H, Bertrand S, development
D'Aniello S

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
