## [Decision Letter]

**Acceptance summary:**

This paper will be of interest to developmental scientists, particularly those interested in the evolution of developmental mechanisms, as well as signalling mechanisms. Through functional and quantitative studies, it reveals a new link between two signalling molecules, nitric oxide (NO) and retinoic acid (RA), in the development of the lancelet pharynx.

**Decision letter after peer review:**

Thank you for submitting your article "Crosstalk between Nitric Oxide and Retinoic Acid pathways is essential for amphioxus pharynx development" for consideration by *eLife*. Your article has been reviewed by 3 peer reviewers, and the evaluation has been overseen by a Reviewing Editor and Marianne Bronner as the Senior Editor. The following individual involved in review of your submission has agreed to reveal their identity: David Ferrier (Reviewer #2).

The reviewers have discussed the reviews with one another and the Reviewing Editor has drafted this decision to help you prepare a revised submission.

All three reviewers agree that the research question under study, the requirement of the cross-talk between two important developmental signaling pathways- retinoic acid and the NO – for amphioxus pharynx development, is in principle interesting and could be suitable for publication in *eLife*.

However, at present there are major open concerns especially on the lack of statistical analyses, quality of data presentation and inconsistencies with previously published work, that need to be addressed.

Although it is the current policy of *eLife* to avoid additional experiments in revisions as much as possible, this is unfortunately likely impossible to fulfill with the current manuscript in order to bring it to a level that matches the standards of *eLife*. However, we think that in many cases an improvement of analyses and data presentation will likely already significantly improve the manuscript.

1) The presented study is a follow-up on a previous paper by the same lab (Annona et al., 2017; DOI:10.1038/s41598-017-08157-w). When comparing the work of this previous study with the current manuscript two major discrepancies are apparent:

– In Annona et al. the two drugs were used to inhibit NOS production: L-NAME and TRIM, while only one inhibitor was used in the present study. Furthermore, there appear to be discrepancies concerning the developmental time windows during which chemical disruption of NO signaling is effective described in the two publications. This needs to be clarified.

– The timing of NosA,B,C expression, the suggested regulation of NosA and B by retinoic acid (RA) and the detected presumptive RARE regulatory elements in the genome don't match. More specifically, NosA,B,C expression at 24 hours (or around this time point) was investigated by Annona et al., 2017. Based on these data, NosA is not expressed during development, whereas NosB and NosC are expressed. In the submitted manuscript, the authors show that NosA and NosB are upregulated upon RA treatment, whereas NosC shows no changes in expression. They therefore suggest that RA regulates NosA and NosB transcription. Since only NosB is expressed during the relevant timepoints at early development, the transcription of this gene could be under the regulation of RA. However, when the authors look into the retinoic acid response elements (RARE) in the genomic region of NosB, they only find a DR3, which is not the typical RARE. They find DR1 and DR5 (apart from DR3's), which are more typical RARE's, in the genomic region of NosA, but as mentioned this gene is not expressed during development. This makes the hypothesis of a direct regulation of NosA and NosB by RA during normal development unconvincing.

Can the author dissolve these apparent discrepancies?

2) The authors study the open chomatin structure at 8, 15, 36 and 60 hours, thus time points, which do not overlap with the drug treatment period (24-30 hours). They need to analyze the genome architecture at this time period.

3) The previous work by Annona et al. 2017 et al., shows that a major peak at NO levels occurs later than the chosen treatment window. How do NO levels during the time window of the experiment compare with other studies, i.e. is there evidence these are relevant levels? This is particularly noteworthy, as there is no control experiment showing that TRIM incubation affects NO levels or NO signaling during the incubation period (e.g. DAF-FM-DA staining or by NO quantification). It is therefore not possible to estimate the specificity of the resulting phenotypes.

We thus request from the authors to provide ISH patterns of all the Nos genes, as well as NO localisation from at least 2 timepoints (e.g. start and end of window) of the TRIM application window.

4) One overarching critique is that the general description of the figures and hence also the phenotypes are of poor quality. An improvement of this point will already majorly improve the entire manuscript.

– Figure 1A: Indicate developmental stages (N2, N4, T1, T2, T3, L0) together with the hours-post-fertilization (hpf) to facilitate the understanding of the treatment period with respect to the development of amphioxus.

– Figure 1B: Outline pharyngeal region e.g. with thin, dashed white lines in longitudinal and cross-sections and indicate relevant anatomical structures (club-shaped gland, endostyle, gill slits) e.g. with an arrow. Is the endostyle positioned more ventrally in TRIM treated larva?

Figure 1C: why are Cyp26.3, Rdh11/12.18 and Crabp shown in triplicates?

– Figure 1B: The 'digital sectioning' method using confocal imaging and reconstruction of nuclear stainings is not suited to characterize the phenotype. Due to the loss of signal in deeper regions, morphological structures (e.g. differences in pharyngeal and gill slit morphology, endostyl, club-shaped glands) are impossible to recognize.

– Figure 3B: the heads of these amphioxus should be annotated to indicate key structures for non-amphioxus specialists. Ideally the images should be higher magnification and resolution as well, as the morphology is currently not very clear.

– Figure 3A and B: Furthermore, the morphological differences between 'altered', 'partially recovered' and 'recovered' is unclear. Figure 3B does not help understanding changes as the pictures are too small to recognize any morphological details without staining, and no structures are indicated. It is also unclear how animals scored as 'altered', 'partially recovered' and 'recovered' differ in their morphological structures. And does 'recovered' mean that these embryos show an initial phenotype that then 'recovers' during development, or do they show a completely normal development?

5) Missing statistics/statistical information:

– Lines 85-89 (Figure 1): Where is the evidence that there is reduction in pharynx length? Where is the evidence for a smaller first gill slit? Measurements with a decent sample size and a basic statistical test must be provided.

– The description of ISH pictures in Figure 2A lacks any quantification and thus any information on the penetrance of the respective phenotypes are (as in Figure 3C). The lack of any 'negative control genes' (the large set of genes that, based on the RNASeq dataset, should not be affected) make it difficult to judge how specific changes in AP axis and RA pathway genes are.

– How did the authors obtain the qRT-PCR calculations? They need to clarify how they obtained the Fold changes shown in the histograms.e.g. by showing the maths behind the result when marking the cells in the excel sheet. The raw data for rpl32 is missing for Crabp in Figure 2B. The qPCR results in Figure 2B-E lack significance tests.

6) The RNA-Seq study needs improvements:

– The PCA (Figure S1C) shows no concordance among control samples or treated samples. Also, the histogram shows a clustering of replicates, and NOT of 'treated' and 'control' samples. This casts doubts on the quality and validity of the RNASeq dataset. These doubts are not removed by the current validation experiments, as these experiments tested only significantly upregulated genes by RNA-Seq, while downregulated and non-significant genes as 'controls' are missing. These additional controls are necessary to assess the validity of the RNA-Seq data.

7) More information about the details of the ATACseq and ChIPseq data used, as well as the general RA responsive elements prediction is required.

– For example, in what amphioxus samples (and treatments if any) are these ATACseq and ChIPseq signals seen? There is some detail provided in the Methods section, but something is odd here and perhaps needs some further explanation. Since the two relevant Nos genes are supposedly not active during development then why do they have ATACseq and ChIPseq signals from embryo and larval samples? Why should these two Nos genes have apparently active regulatory elements focused on RAREs when the genes are not normally expressed under the control of RA, but only become active when exogenous RA is applied? We may well have missed something in the logic here, but this merely shows that the current level of explanation is insufficient.

– The analysis of RA responsive elements lacks statistical analysis and depth. It is left unclear how many RAREs would be expected by chance on a 52kb resp. 25kb locus. In addition, the authors include all ATAC-Seq peaks from stages ranging between 8h and 60hpf, while the window of RA responsiveness has been tightly restricted to the 24h-30hpf window. Also, as NosC expression levels stay constant upon RA incubation, it would be crucial to know if the NosC locus lacks any open RARE sites (as would be expected).

– The authors use NHR-SCAN tool to predict putative direct repeats binding sites in the genomic sequence of NosA and NosB. Which consensus sequence does the program follow? It appears that it does not follow the consensus sequence for typical RARE ((A/G)G(G/T)TCA), since the sequence for DR1 deviate from this sequence?

DR1, DR2 and DR5 are the commonly described binding RARE's for the RAR/RXR heterodimers. Further, DR8 has been described as retinoic acid dependent regulation of gene transcription through RAR/RXR (Moutier et al., 2012). The authors need to provide clarification which are the most commonly used RARE's of the DR's detected.

– Please also mention if RAREs fall within an intron in the genomic regions of the Nos genes, since the transcriptional regulation through RARE is often associated to introns.

8) Information on the concentration dependency of compounds used in the rescue experiment is lacking. Please explain why the BMS009 concentration used here (10exp-6 M) is 10x higher than the highest concentration used in the original publication on amphioxus pharynx development (Escriva et al., Development 2002).

9) There are multiple cases of incorrect labels/statements, which give the impression of a lack of vigorous cross-checking of the manuscript. This must be improved:

– Figure 1A: Red/green labels are swapped for wildtype phenotype (green) vs altered phenotype (red).

– Figure 1A: Line 92: 24 hours should be 18 hours.

– Line 96: the critical time for NO action, judging from the TRIM treatments, is from 24-42 hpf, as 6-hour treatment windows spread throughout this period give a high level of altered phenotypes. The statement on line 96 is thus not correct.

The white bar in Figure 2D looks like an attempt to show that y-axis values between 5 and some other non-indicated value have been omitted. Clarify what it is exactly supposed to show and do not draw it over the error bars.

– Line 254:

Correct to "important to improve"

– Figure 1 figure legend: in vivo in italic.

– Line 100: remove dot after Figure

– Line 105: remove space after Figure 1

– Line 135: write iii. instead of iii)

– Figure legend Figure supplement 1:

C. should be C)

Blu should be Blue

– Figure 2A:

To be consistent with the remaining figures, please indicate arrowheads at the posterior position of Cyp26.2 – even though no change in AP expression is observed.

Figure legend for Figure 2:

For clarity, I would suggest to define the drug treatment window for B-E: "…after 6 hours of pharmacological TRIM or RA treatments (24-30hpf) of:…"

pvalue should be p-value.

– figure supplement 3:

Panel B:

Write NosA in italic

Figure legend to Figure supplement 3:

pvalue should be p-value

– Line 201:

Dot is underlined. Remove underline.

– Line 306-307 and line 358-359:

Correct link to references.

– Line: 319-320:

Indicate which retinoic acid isoform was used for the study.

– Line 391-392:

Indicate type of t-test and data entry format.

10) A summary drawing of the regulatory loop between NO and RA would be informative, also indicating the known target genes (from this study).

[Editors' note: further revisions were suggested prior to acceptance, as described below.]

Thank you for submitting your article "Crosstalk between Nitric Oxide and Retinoic Acid pathways is essential for amphioxus pharynx development" for consideration by *eLife*. Your article has been reviewed by 3 peer reviewers, and the evaluation has been overseen by a Reviewing Editor and Marianne Bronner as the Senior Editor. The following individuals involved in review of your submission have agreed to reveal their identity: Mette Handberg-Thorsager (Reviewer #1); David Ferrier (Reviewer #2).

All three reviewers agreed that the revisions have substantially improved the manuscript. The extent of additional experiments, even including the establishment of a new technique for NO measurements in amphioxus (DAN measurements) is very positively remarkable. Furthermore, the rescue experiments in which the authors reduced RA signaling alongside inhibiting NO signaling provides a very important confirmation of the NO-RA signaling pathways crosstalk.

However, there is still some disagreement among the reviewers, if all comments were sufficiently addressed. It might be that the remaining concerns are simply due to insufficient explanations. Thus, we agreed that no additional experiments are required, but we ask the authors to very carefully and in detail address the remaining concerns raised below and where necessary adjust the manuscript accordingly:

Essential revisions:

1) The authors contradict themselves in the expression of the NosB gene. They state in the present author response letter: "Nevertheless, we confirmed that NosA and NosB are not expressed at these stages…." This is a direct contradiction to Figure 2 of Annona, G et al., 2017.

The reviewers and RE are aware that further research into a topic can lead to the realization that previously published data (even if acquired with all best practice and controls) are incomplete or even incorrect. However, if this happens then this needs to be transparently clarified in the following work.

2) Depending on what now really is the case with the expression of NosB, a further discussion by the authors on how the chosen TRIM drug treatment period (24-30 hpf) makes biological sense is required.

– Under normal conditions, NosA is not expressed during the developmental period shown to be affected by an inactivation of the NOS protein by TRIM incubation. NosB is also not expressed or only very weakly. This suggests either that low levels of the NOS protein are sufficient to produce the right amount of NO at 24 hpf or that the turnover of the NOS protein is slow and still present at 24 hpf (although translated at earlier stages, where NosB expression is higher (shown for 10 hpf in Annona et al., 2017)). NO itself is probably not stable enough to stay in the tissue. The authors should discuss this point.

– NosA and -B expression increases after RA incubation at developmental stages, where the genes are normally not expressed or only lowly expressed. Does this mean that a possible regulation, direct or indirect, of Nos expression by RA is not occurring under normal developmental conditions? Could it indicate that this part of the regulatory loop potentially could be used at other developmental stages? – e.g. regulating NosB expression during blastula/early neurula stages or regulating NosA expression in adults (according to the expression levels published in Annona et al., 2017)? Or is this part of the regulatory loop only activated when an NO/RA unbalance occurs? NosC genomic region has many RARE's, but is not regulated by RA at the developmental time, which the authors have analyses. Maybe NosC could be regulated by RA earlier or later in development? Again, the authors should discuss this point.

3) Related to the point 2 above- In answer to the discrepancy between the previous Annona paper and the current manuscript with regards to the differing developmental windows that are sensitive to TRIM treatments, the authors have explained in their response why this is, but not clearly described this in the revised manuscript. It is important to make it clear to readers that the currently reported windows are the correct ones. Without such an explicit statement that the previous Annona work is incorrect in this regard, these important details will be confusing to readers who are trying to follow the work from this lab. It has clearly confused the referees.

4) RAREs: The authors state that "Moreover, we searched in a number of genomic fragments (50 kb long) randomly chosen and found a comparable number of putative RAREs to Nos genes loci. »

This makes it very clear that there is no significant enrichment of RAREs in the Nos genes and hence the data must not be presented as support for a regulation by RA. So if the authors choose to still include this observation, it has to be presented as it is: as non-significant. For this reason, a statement such as 'RA regulates the transcription of Nos genes, probably through RA Response Elements found in their regulatory regions' (abstract) has to be removed and the according sections in the main text re-written or removed.

5) Figure 1 —figure supplement 1A (+ lines 90-91):

– Here, the effect of TRIM is evaluated by measuring the pharyngeal length. Optimally, the total body length should also be taken into account, so the comparison would be of the pharyngeal length/the total body length. What if the larvae are overall shorter, then maybe there is no difference in the pharyngeal length with respect to the total body length in the TRIM treated larvae (but yes in the morphology and position of the pharyngeal structures).

6) In Figure 1-Supplement 1B: Clearly indicate at which time point the CTRL was taken. Furthermore, the most appropriate way to perform this experiment would be to take a control sample at the beginning and the end of each the different time windows of treatment. Ideally you could add this. If not, please carefully explain why this was not done and why the data are nevertheless meaningful.

7) Figure 1B: is the scale of these reconstructions the same between panels? It looks as though it is not, and that the TRIM-treated images might have been slightly enlarged relative to the controls. If this is the case, the authors need to change this so that everything is at the same scale. As currently presented, it is difficult for readers to appreciate the shortening of the pharynx region in TRIM-treated larvae (it certainly does not look like 40% shortening in Figure 1B).

8) Line 105-106: The authors claim that the 'critical time of its [NO] action is restricted approximately to the first six hours (24-30hpf)'. Given that incubations starting at 36h still result in 75% defective embryos, this claim is factually wrong.

9) Figure 1-supplement 4: is it correct that the authors compared Cdx, Six1/2 and IrxC in the RNAseq and RT-PCR panel, but then Cdx, Six1/2, Pitx in the ISH panel, or should this be the same three genes in both panels?

10) With the new inclusion of genes that are supposedly not regulated by NO/RA in the stages of development examined in this revised manuscript, something striking has become apparent. Is NO inhibition with TRIM supposed to be posteriorizing the embryos as a whole, or in a more restricted fashion (either in space or time)? If it were a general posteriorization, why then do Cdx, Pitx and Six1/2 not shift? Particularly since amphioxus Cdx is a gene that has previously been shown to be sensitive to changes in RA levels (Osborne et al., 2009, Developmental Biology 327: 252-262). Is this simply due to not enough time elapsing between the TRIM treatments performed here, the response of the first suite of genes involved in RA signaling, but then not the next tranche of responsive genes downstream of RA? This, however, does not seem to be the entire (or even likely) explanation, as the authors RA manipulations supposedly phenocopy the TRIM treatments.

Instead, do the different effects of manipulating RA levels in the current study relative to previous studies reflect the different developmental times that RA levels are altered between studies, thus producing shifts in A-P regionalised genes to different extents or in different regions between different studies? Perhaps this relates to the temporal shifting of the RA production centre as amphioxus development proceeds, this shifting process already being briefly mentioned by the authors, and this could act in conjunction with the NO signaling centre not shifting but remaining in the pharyngeal region (if this is the case)?

In this context, the comment on line 225 about "posteriorization of the larval body plan" is misleading if the shift is not actually in the entire body plan but only in the pharyngeal region.

These observations warrant a brief discussion of A-P patterning in amphioxus to clarify more precisely how the authors think the phenotypes they observe are generated in the context of amphioxus development more widely, and why their phenotype seems to be restricted to the pharynx region when RA-linked phenotypes are much more widespread in amphioxus.

Further suggestions for improvement:

Figure 4B:

A modification of the summary figure (Figure 4) would facilitate the reading of this figure. Now a decrease or an increase of a molecule (NO, RA) or a gene product is indicated in many ways: thickness of the arrows between two components, the size of the molecule or gene product and the addition of red or green arrows to show a decrease or an increase, respectively. The authors need to make this figure clearer e.g. through 1) showing all connecting arrows in one size and 2) use only one type of arrows, so no arrows with dashed lines (or explain better in the figure legend). A distinction in arrows could be based on the process (enzymatic process (Nos -> NO and Aldh1a -> RA) vs transcriptional regulation (NO -> Aldh1a, RA -> Nos and RA -> genes of the RA machinery)). 3) Choose either to show a decrease or an increase by the size of the molecules and gene products OR by adding the green (increase) and red (decrease) arrows, but not both. It's to much information. 4) Remove the thick arrow between Figure 4B and 4C.

– The authors need to indicate what causes the decrease in NO levels in Figure 4B e.g. by placing a cross over the arrow between Nos and NO to illustrate an inactivation of Nos.

– add a question marker to the arrow between NO and Aldh1a to indicate that it is not known exactly how this regulation takes place.

– instead of "RA levels reduction" maybe write "retinoid metabolism". That RA levels are reduced is indicated with the arrow.

– For consistency, the authors could write "Normal state" over Figure 4C as for Figure 4A, or "Recovery of normal state".

line 87:

– regarding the treatment window from N2 to T2. Shouldn't this be N2 to T3?

line 90-94:

– revise here all references to the figure e.g. in line 92 and 93, the reference should be Figure 1B panel I-II and V-VI (and not Figure 1B, panel I-II and III-IV). References to part of the figure 1B are missing in the text (Figure 1B III, IV, VIII). The authors should include these e.g. referring to the other matching panels, thus refer to panel I-III, II-IV, V-VII and V-VIII together, respectively.

line 99:

– add figure reference to Figure 1A after "…70% affected larvae" to facilitate reading.

line 101:

The authors write that 'NO levels ….dropped to 40% in comparison to controls'. As the '40%' is based on comparing average concentrations, the authors should report a range rather than an exact number.

line 135:

(Figure 2A, D) and not "(Figure 2A, B)".

line 194:

correct to "described"

line 322-324:

– Add "shown indirectly" to be clear: "A reduction of NO (Figure 4B), as shown by our results, induces an excess of RA, shown indirectly through the overexpression of Aldh1a."

line 346 Key Resources Table:

– remove the column "additional information" if there are no additional information.

line 660: write Cyp26.2 in italic

Figure 1 —figure supplement 1A.

– length not "lenght" on the y-axis

There is still some further editing and proofing of the written English required. There are a handful of odd word choices (e.g. change sensible/insensible to sensitive/insensitive) and slightly awkward sentence constructions.

[Editors’ note: further revisions were suggested prior to acceptance, as described below.]

Thank you for submitting your article "Crosstalk between Nitric Oxide and Retinoic Acid pathways is essential for amphioxus pharynx development" for consideration by *eLife*. Your article has been reviewed by 3 peer reviewers, and the evaluation has been overseen by a Reviewing Editor and Marianne Bronner as the Senior Editor. The following individuals involved in review of your submission have agreed to reveal their identity: Mette Handberg-Thorsager (Reviewer #1); David Ferrier (Reviewer #2).

All reviewers agree that the authors have sufficiently addressed their major concerns and that no further full cycle of revisions is necessary. However, reviewers 1 and 3 very constructively point out several aspects that we would ask the authors to carefully consider for implementation when preparing the final version of their manuscript.

*Reviewer #1:*

Recommendations for the authors on the revised manuscript:

The manuscript reads very well now, but I think some of their main suggestions and discussions have to be revised to support the main message of the manuscript. Here I'm referring to the following observations and statements by the authors:

1. NosC is responsible for the production of NO from N2 to L1 larval stage (line 101-103).

2. RA and TRIM treatment experiments cause an increase in NosA and NosB expression, but not in NosC (line 194-205, Figure 2E) – why they conclude:

"A possible explanation could be that while in normal conditions RA does not regulate NosA and NosB expression, it directly or indirectly regulates such gene expression in the case of a NO/RA inbalance as a way to restore the correct NO/RA ratio."(line 343-345)

Based on these conclusions, the title of the manuscript "Crosstalk between Nitric Oxide and Retinoic Acid pathways is essential for amphioxus pharynx development" is not physiological relevant. According to the authors, the crosstalk only occurs when an NO/RA unbalance takes place. To me, this is a problem.

I think the authors are missing to include another plausible possibility, which would give biological meaning to the work and which they actually do discuss in the Response to Review Comments: NosB could also contribute to NO production during normal pharynx development (this could still be together with NosC). The results from Annona et al. 2017 supports this: Expression levels validated by ddPCR of NosA, NosB and NosC are shown in Figure 2h-h'. When expressed, both NosA and NosB show ~20x higher expression levels than NosC – also when comparing expression levels between NosB and NosC at 24 hpf, which is when they initiated the 6 hours TRIM treatment. Now of course, it would be interesting to have the in situ hybridisation for NosB at 24 hpf. In Annona et al., 2017 they show the expression pattern until mid-gastrula stage and not during the Nerula period (TRIM treatment starts at N2). In the Response to Review Comments, the authors declare that they did the in situ hybridisation for NosB at 24 hpf and didn't detect any pattern. They argue that this could be because of a broader weaker expression. This makes sense. I think they should mention this in the manuscript. And also that the NosB expression during mid-gastrula was a broad expression in the ectoderm, next to the endoderm (where I guess the pharyngeal tissues develop from). Including the possibility that NosB could contribute to NO production during pharynx development would give a biological meaning to the conclusion that there is a crosstalk between NO and RA pathways during amphioxus pharynx development. And not that the crosstalk only takes place during a NO/RA inbalance relying on the activity of NosA and NosB, which are normally not expressed during pharynx development (as stated in the current version of the manuscript, but only partially true as mentioned above).

In this regard, maybe a short discussion on NO production site (based on NosB and NosC expression) vs NO target tissues (Aldh1a expression) could help clarify the above concern.

Reviewer #2:

I am happy to now endorse publication of this manuscript. The authors have addressed the previous queries that I had, and I thank them for their engagement with this process.

Reviewer #3:

The authors have addressed several key weaknesses of the manuscript:

- The confusion and discrepancies between the Annona et al. 2017 paper and the current manuscript have been satisfactorily addressed.

- The authors have removed a non-significant in silico prediction of potential RARE binding sites from the abstract, results and discussion.

- They have also clarified the discrepancies between different NO inhibitors between the Annona et al. 2017 paper and the current manuscript.

---

## [Author Response]

1) The presented study is a follow-up on a previous paper by the same lab (Annona et al. 2017 ; DOI:10.1038/s41598-017-08157-w). When comparing the work of this previous study with the current manuscript two major discrepancies are apparent:– In Annona et al. the two drugs were used to inhibit NOS production: L-NAME and TRIM, while only one inhibitor was used in the present study. Furthermore, there appear to be discrepancies concerning the developmental time windows during which chemical disruption of NO signaling is effective described in the two publications. This needs to be clarified.

We thank the reviewers to raise this point and will try to clarify it.

1. In our previous publication, Annona et al., (2017), we tested the efficiency and toxicity of two different inhibitors of NOS activity: L-NAME (an analog of arginine that is a NOS substrate) at 1 mM (best concentration) gave poor consistent results, while TRIM at 100 μm (best concentration) gave a higher rate of reproducible phenotypes. We observed this during several amphioxus spawning seasons, and therefore decided to invest our efforts on TRIM treatments in order to uncover the characterization of the developmental mechanism downstream of NO.

2. Our previous conclusion about the putative temporal window in which NO acts (Annona et al., 2017) was based on preliminary experiments that where not reported in Results section, but in a single sentence in the discussion. During the following amphioxus spawning seasons the short-term TRIM treatments were performed in a systematically manner on several batch of embryos and using different stocks of the drug. This allowed us to be more confident about the obtained results to which we dedicated more work and on which we decided to build the RNAseq experiments and analysis.

– The timing of NosA,B,C expression, the suggested regulation of NosA and B by retinoic acid (RA) and the detected presumptive RARE regulatory elements in the genome don't match. More specifically, NosA,B,C expression at 24 hours (or around this time point) was investigated by Annona et al., 2017. Based on these data, NosA is not expressed during development, whereas NosB and NosC are expressed. In the submitted manuscript, the authors show that NosA and NosB are upregulated upon RA treatment, whereas NosC shows no changes in expression. They therefore suggest that RA regulates NosA and NosB transcription. Since only NosB is expressed during the relevant timepoints at early development, the transcription of this gene could be under the regulation of RA. However, when the authors look into the retinoic acid response elements (RARE) in the genomic region of NosB, they only find a DR3, which is not the typical RARE. They find DR1 and DR5 (apart from DR3's), which are more typical RARE's, in the genomic region of NosA, but as mentioned this gene is not expressed during development. This make thehypothesis of a direct regulation of NosA and NosB by RA during normal development unconvincing.Can the author dissolve these apparent discrepancies?

We agree that this part was not clear enough and we improved it according to the reviewer’s recommendations. First, regarding the atypical RAREs found in the previous version, we realized that several RAREs were indicated in reverse strand, therefore not recognizable at first sight, while others were of unconventional types, although possibly real RAREs. To improve this and to be on the safe side, we now show exclusively typical DR RAREs, predicted with NHscan using a DR state of 0,005.

Moreover, we now also include the analysis of NosC locus that we intentionally did not include previously because the level of expression of this gene was unchanged after RA treatment. The output is that we also found several RAREs in the NosC locus that are now presented in Figure 2—figure supplement 1B. Despite this, functional experiments with RA addition show that, unlike the other two genes, NosC is not regulated by RA.

In addition, the message of this study is that RA and NO signals are finely tuned to each other. In other words, the fact that a gene (i.e. NosA) is not expressed under normal conditions during development does not mean that in extraordinary situations where an excess of RA may generate a lethal phenotype, this gene cannot be over-expressed to compensate for the excess of RA. On the contrary, the presence of RAREs that putatively control the expression of the Nos genes (whether they are expressed or not in normal conditions), goes in this direction of strengthening the cross regulation of these two signals so that the level of them is the right level and that the phenotype of the larva is viable

Our decision to include DR3 as a RARE is supported by previous work, as for example: “Retinoic acid response element in HOXA-7 regulatory region affects the rate, not the formation of anterior boundary expression. Kim MH et al., Int J Dev Biol. 2002 May;46(3):325-8. PMID: 12068955.”

2) The authors study the open chomatin structure at 8, 15, 36 and 60 hours, thus time points, which do not overlap with the drug treatment period (24-30 hours). They need to analyze the genome architecture at this time period.

We understand that it might have been unclear why we used ATAC-seq data. Our aim by looking at ATAC-seq and ChIP-seq data was to define open chromatin regions during development in the vicinity of Nos genes in order to reduce the search for RARE to putative enhancer regions. Even if the data available do not correspond to the developmental stages studied in detail in our work, they cover the main developmental stages and hence give us the opportunity to define at a large-scale open chromatin regions. If we would have undertaken new ATAC-seq experiments, which is out of the scope of this study, we would have looked not only at Nos genes locus but would have undertaken a whole genomic analysis which was not our objective here. However, we agree that such experiment would be extremely interesting and could be the focus of future studies.

3) The previous work by Annona et al. 2017 et al., shows that a major peak at NO levels occurs later than the chosen treatment window. How do NO levels during the time window of the experiment compare with other studies, i.e. is there evidence these are relevant levels? This is particularly noteworthy, as there is no control experiment showing that TRIM incubation affects NO levels or NO signaling during the incubation period (e.g. DAF-FM-DA staining or by NO quantification). It is therefore not possible to estimate the specificity of the resulting phenotypes.We thus request from the authors to provide ISH patterns of all the Nos genes, as well as NO localisation from at least 2 timepoints (e.g. start and end of window) of the TRIM application window.

The fact that the pick of NO level was previously observed in larva does not mean that NO only acts at this stage. Indeed, it is well documented that NO acts at low concentrations. However, we thank the reviewers for this comment and we undertook new experiments in order to demonstrate that TRIM affects NO levels. We used DAN assay for NO quantification in control and TRIM-treated embryos. DAN assay is more sensitive than the Griess assay previously used by us and other authors. Data analysis from TRIM treatments at 24-30 hpf, 24-36 hpf, and 24-42 hpf highlighted that TRIM-treated embryos have a 60% reduction of endogenous NO compared to controls.

We included the paragraph:

“Fluorimetric determination of endogenous NO concentration” in the methods section and added it in the first paragraph of Results as follow: “Endogenous NO levels tested after TRIM incubation in following intervals 24 to 30 hpf, 24 to 36 hpf and 24 to 42 hpf, dropped to 40% in comparison to controls (Figure 1—figure supplement 1B).”

Moreover, as request we performed ISH of Nos genes at the two time points (24 and 42 hpf). Nevertheless, we confirmed that NosA and NosB are not expressed at these stages and the increase of expression after TRIM treatment was not sufficient to be appreciated by ISH, since it is not a quantitative technique. For NosC expression pattern we did not appreciate any difference in controls and TRIM-treated embryos, and in conclusion the pattern is identical to the one previously published (Annona et al., 2017). For this reason we do not think necessary to show these findings in a new figure of the manuscript.

4) One overarching critique is that the general description of the figures and hence also the phenotypes are of poor quality. An improvement of this point will already majorly improve the entire manuscript.

We have greatly improved the figures presenting the phenotypes. In Figure 1, we now show the different pharyngeal structures of larva after 3D reconstruction from confocal nuclei imaging and we have highlighted the pharyngeal structures in colour in Figure 3. We hope that now the phenotypes can be clearly visualized even for non-specialists of the amphioxus animal model.

– Figure 1A: Indicate developmental stages (N2, N4, T1, T2, T3, L0) together with the hours-post-fertilization (hpf) to facilitate the understanding of the treatment period with respect to the development of amphioxus.

We agree and improved the figures by using a modern nomenclature stage system that was recently published:

“An updated staging system for cephalochordate development: one table suits them all by Carvalho et al., bioRxiv 2020.05.26.112193”.

We also added the following reference to the Materials and methods section.

– Figure 1B: Outline pharyngeal region e.g. with thin, dashed white lines in longitudinal and cross-sections and indicate relevant anatomical structures (club-shaped gland, endostyle, gill slits) e.g. with an arrow. Is the endostyle positioned more ventrally in TRIM treated larva?

We now indicate relevant anatomical structures with different colours in the 3D reconstruction: purple = pre-oral pit, red = mouth, light blue = endostyle, yellow = club-shaped gland, green = gill slit.

We noticed thanks to these reconstructions that, as suggested by the reviewer, the endostyle is more ventral in TRIM-treated larva than in controls. We hence included this point in the description of the phenotype in the Results paragraph “NO controls pharyngeal development during early neurulation in amphioxus”, as follows: “…iii. an incomplete formation of the club-shaped gland and of the endostyle, the latter being positioned more ventrally than in controls (Figure 1B)”.

Figure 1C: why are Cyp26.3, Rdh11/12.18 and Crabp shown in triplicates?

We would like to thank the reviewer for this comment. It was indeed a mistake that has been corrected in this version.

– Figure 1B: The 'digital sectioning' method using confocal imaging and reconstruction of nuclear stainings is not suited to characterize the phenotype. Due to the loss of signal in deeper regions, morphological structures (e.g. differences in pharyngeal and gill slit morphology, endostyl, club-shaped glands) are impossible to recognize.

Taking advantage of the previous Z-stack confocal acquisitions, we performed a 3D reconstruction in which we highlighted the different anatomical structures of interest using a colour code. We agree that the present representation is better suited to characterize the phenotype of TRIM-treated larvae. We modified this point in the text, methods and figure 1 legend, accordingly.

– Figure 3B: the heads of these amphioxus should be annotated to indicate key structures for non-amphioxus specialists. Ideally the images should be higher magnification and resolution as well, as the morphology is currently not very clear.

We improved substantially Figure 3B, as suggested.

– Figure 3A and B: Furthermore, the morphological differences between 'altered', 'partially recovered' and 'recovered' is unclear. Figure 3B does not help understanding changes as the pictures are too small to recognize any morphological details without staining, and no structures are indicated. It is also unclear how animals scored as 'altered', 'partially recovered' and 'recovered' differ in their morphological structures. And does 'recovered' mean that these embryos show an initial phenotype that then 'recovers' during development, or do they show a completely normal development?

As stated above, we improved the figure presentation and now indicate key anatomical structures. Moreover, we hope we made the phenotype classification clearer by describing in the text the phenotypes associated with each category: “rescue”, observation of wild type phenotype; “partial rescue”, observation of an intermediate phenotype with a length and structure organization of the pharynx similar to control, but with a smaller mouth compared to wild-type larva; “altered”, having an affected phenotype as descried in TRIM-treated larva.

Regarding the recovery of the development, we observed a recovery of the normal morphology at the larva stage as a consequence of the fact that embryos follow a completely normal development as demonstrated by the in situ hybridization of Figure 3C.

5) Missing statistics/statistical information:– Lines 85-89 (Figure 1): Where is the evidence that there is reduction in pharynx length? Where is the evidence for a smaller first gill slit? Measurements with a decent sample size and a basic statistical test must be provided.

We systematically measured the length of the pharynx in 20 wild type larvae and 20 TRIM-treated larvae. As shown in Figure 1—figure supplement 1A we observed a reduction of the pharynx length of about 40%, significantly supported by the paired t-test statistical analysis. These data are now included in the text in the first paragraph of the Results section. We described how measurements were carried out at the end of the Pharmacological treatment paragraph in the Materials and methods section.

Moreover, thanks to the 3D reconstruction we realized that the first gill slit is not smaller in TRIM treated larvae as we declared, so we eliminate this from the description of the phenotype in the first paragraph of the Results.

– The description of ISH pictures in Figure 2A lacks any quantification and thus any information on the penetrance of the respective phenotypes are (as in Figure 3C). The lack of any 'negative control genes' (the large set of genes that, based on the RNASeq dataset, should not be affected) make it difficult to judge how specific changes in AP axis and RA pathway genes are.

We added the request information in the Figure 2A legend. The penetrance of the signal is of 100%. We used 15 embryos/larvae for each probes, all of them presenting the same pattern. Although it is not common to test the expression of “negative control genes” in the context of validation of RNA-seq data, we included as asked by the reviewer the ISH of three negative control genes: Six1/2, Cdx, Pitx in Figure 1—figure supplement 4D.

– How did the authors obtain the qRT-PCR calculations? They need to clarify how they obtained the Fold changes shown in the histograms.e.g. by showing the maths behind the result when marking the cells in the excel sheet. The raw data for rpl32 is missing for Crabp in Figure 2B. The qPCR results in Figure 2B-E lack significance tests.

For the qRT-PCR calculations we used the 2−ΔΔCt method. We added this information in the Materials and methods section. Moreover, we converted the Source data 2 file in a calculation sheet (exel).

We added the Rpl32 row data for Crabp in Source data 2 file (relative to Figure 2B).

For the qRT-PCR results we added the information on statistical significance in Figure 2B-E and in the legend: The statistical significance is indicated: * = p-value<0,05; ** = p-value<0,01; **** = p-value<0,0001.

6) The RNA-Seq study needs improvements:– The PCA (Figure S1C) shows no concordance among control samples or treated samples. Also, the histogram shows a clustering of replicates, and NOT of 'treated' and 'control' samples. This casts doubts on the quality and validity of the RNASeq dataset. These doubts are not removed by the current validation experiments, as these experiments tested only significantly upregulated genes by RNA-Seq, while downregulated and non-significant genes as 'controls' are missing. These additional controls are necessary to assess the validity of the RNA-Seq data.

The PCA (Figure 1—figure supplement 2C) shows a discrimination between "control" samples (in red) and "treated" samples (in green) along the abscissa: red dots are on the left (negative values), and green dots on the right (positive values). This means that the treatment condition is the main principal component (PC1), explaining 47% of the variance between all samples.

We agree with the reviewer that the heatmap (Figure 1C, and Figure 1—figure supplement 2B) shows a clustering by replicate (i.e. eggs from a same female) and not by treatment condition. This is due to the particularity of our model: females are directly collected from the field and display a strong genomic variability. Our experiment has been designed to avoid any batch effect: for each biological replicate, the two treatment conditions were applied. DEseq2 takes into account such design when calculating adjusted p-values with Benjamini-Hochberg false discovery correction (Love et al. 2014). In addition, we underlie that, when comparing biological replicates: for a given sample, the samples of the same treatment condition are closer than the samples from the other treatment condition.

Thus, we argue that the validity of our RNAseq dataset is not debatable on this aspect.

*Love, Michael I., Wolfgang Huber, and Simon Anders. "Moderated estimation of fold change and dispersion for RNA-seq data with DESeq2." Genome Biology 15.12 (2014): 550.

However, in order to further validate the RNAseq data, we included the following additional controls: 1. qRT-PCR of downregulated genes: FoxE, RunX, Dmrt and Pdgfr, and 2. qRT-PCR of non affected genes: IrxC, Six1/2 and Cdx. We reported these additional results in Figure 1—figure supplement 4B and C.

7) More information about the details of the ATACseq and ChIPseq data used, as well as the general RA responsive elements prediction is required.– For example, in what amphioxus samples (and treatments if any) are these ATACseq and ChIPseq signals seen? There is some detail provided in the Methods section, but something is odd here and perhaps needs some further explanation. Since the two relevant Nos genes are supposedly not active during development then why do they have ATACseq and ChIPseq signals from embryo and larval samples? Why should these two Nos genes have apparently active regulatory elements focused on RAREs when the genes are not normally expressed under the control of RA, but only become active when exogenous RA is applied? We may well have missed something in the logic here, but this merely shows that the current level of explanation is insufficient.

We now included in M and M information useful to data access:

“We used ATAC-seq and ChIP-seq data already published by Marlétaz et al., 2018 (available in http://amphiencode.github.io/).”

Moreover, it should be noted here that ATAC-seq peaks, are usually seen as “active regions”, using the term "active" as a region that positively regulates the expression of the nearby gene. However, gene expression regulation depends on a complete regulatory landscape, containing from few to many different interactions between the enhancers and the promoter (see Bolt CC, Duboule D. The regulatory landscapes of developmental genes. Development. 2020;147(3):dev171736. doi: 10.1242/dev.171736.).

The identification of “active” chromatin regions (both positive or negative in the control of gene expression) is extremely difficult and the pleiotropic effect of the different enhancers on a given promoter can be as unique as the many genes present in a given genome. For example, we showed recently that in a counterintuitive way, the restriction of an expression pattern of a paralogue after a gene duplication, is produced mainly through an increase and not a decrease of the number of enhancers controlling this expression (Marlétaz et al., 2018 Nature). In other words, the presence of ATAC-seq peaks, and therefore of putative regulatory regions, in the proximity of the Nos genes, which are not expressed during development, suggests that the regulation (lack of expression) of these genes is very important for the correct development of the embryo. In addition, the presence of RAREs in these putative regulatory zones, may mean, as we have explained before, that these genes must be finely regulated against a potentially lethal increase in RA. Of course, this explanation is purely speculative and its demonstration requires an in-depth study of each enhancer and the possible epistatic effects among them for each of the Nos genes. This study, although extremely interesting, is evidently outside the scope of this manuscript.

– The analysis of RA responsive elements lacks statistical analysis and depth. It is left unclear how many RAREs would be expected by chance on a 52kb resp. 25kb locus. In addition, the authors include all ATAC-Seq peaks from stages ranging between 8h and 60hpf, while the window of RA responsiveness has been tightly restricted to the 24h-30hpf window. Also, as NosC expression levels stay constant upon RA incubation, it would be crucial to know if the NosC locus lacks any open RARE sites (as would be expected).

As stated in the answer to previous comment, the analysis of RAREs in the NosC locus is now included.

Regarding the time window of ATAC peaks, as clarified in the comment 2, they cover the main developmental stages and hence give us the opportunity to define at a large-scale open chromatin regions.

Moreover, we searched in a number of genomic fragments (50 kb long) randomly chosen and found a comparable number of putative RAREs to Nos genes loci. Nevertheless, although speculative, we think that RAREs highlighted in NosA and NosB loci are potentially functional because supported by our experimental evidences.

– The authors use NHR-SCAN tool to predict putative direct repeats binding sites in the genomic sequence of NosA and NosB. Which consensus sequence does the program follow? It appears that it does not follow the consensus sequence for typical RARE ((A/G)G(G/T)TCA), since the sequence for DR1 deviate from this sequence?DR1, DR2 and DR5 are the commonly described binding RARE's for the RAR/RXR heterodimers. Further, DR8 has been described as retinoic acid dependent regulation of gene transcription through RAR/RXR (Moutier et al., 2012). The authors need to provide clarification which are the most commonly used RARE's of the DR's detected.

We used NHR-SCAN that follow ((A/G)G(G/T)TCA) consensus sequence and, as said above, some of DR sequences were reported in the first version of the manuscript in the opposite direction. We corrected this now.

We agree that the commonly described binding RARE's for the RAR/RXR heterodimers are DR1, DR2, DR3, DR5 and DR8, and therefore we decided to show those exclusively, since experimental proofs for the other DRs is poor.

– Please also mention if RAREs fall within an intron in the genomic regions of the Nos genes, since the transcriptional regulation through RARE is often associated to introns.

We now include the information regarding the exact position of each RARE element (see Figure 2—figure supplement 4B).

8) Information on the concentration dependency of compounds used in the rescue experiment is lacking. Please explain why the BMS009 concentration used here (10exp-6 M) is 10x higher than the highest concentration used in the original publication on amphioxus pharynx development (Escriva et al., Development 2002).

In Escriva et al., (Development, 2002) different concentrations of BMS009 were used (from 10-7 to 10-5 M) with a dose response result. Here, we used 10-6M BSM009 since this was the lowest concentration showing the highest phenotype penetrance.

9) There are multiple cases of incorrect labels/statements, which give the impression of a lack of vigorous cross-checking of the manuscript. This must be improved:

We have checked the whole manuscript and we hope we have now corrected all the mistakes.

– Figure 1A: Red/green labels are swapped for wildtype phenotype (green) vs altered phenotype (red).

We changed this as appropriate.

– Figure 1A: Line 92: 24 hours should be 18 hours.

Done.

– Line 96: the critical time for NO action, judging from the TRIM treatments, is from 24-42 hpf, as 6-hour treatment windows spread throughout this period give a high level of altered phenotypes. The statement on line 96 is thus not correct.vThe white bar in Figure 2D looks like an attempt to show that y-axis values between 5 and some other non-indicated value have been omitted. Clarify what it is exactly supposed to show and do not draw it over the error bars.– Line 254:Correct to "important to improve"

Done.

– Figure 1 figure legend: in vivo in italic.

Done.

– Line 100: remove dot after Figure

Done.

– Line 105: remove space after Figure 1

Done.

– Line 135: write iii. instead of iii)

Done.

– Figure legend Figure supplement 1:C. should be C)Blu should be Blue

Done.

– Figure 2A:To be consistent with the remaining figures, please indicate arrowheads at the posterior position of Cyp26.2 – even though no change in AP expression is observed.

Done.

Figure legend for Figure 2:For clarity, I would suggest to define the drug treatment window for B-E: "…after 6 hours of pharmacological TRIM or RA treatments (24-30hpf) of:…"pvalue should be p-value.

Done.

—figure supplement 3:Panel B:Write NosA in italic

The panel in this figure changed in the new version, anyway we followed the suggestion.

Figure legend to Figure supplement 3:pvalue should be p-value

Done.

– Line 201:Dot is underlined. Remove underline.

Done.

– Line 306-307 and line 358-359:Correct link to references.

Done.

– Line: 319-320:Indicate which retinoic acid isoform was used for the study.

We employed the “all-trans Retinoic Acid”, and added this information in Materials and methods.

– Line 391-392:Indicate type of t-test and data entry format.

We employed the paired t-test for all our analysis: on measurements for the pharynx length, on fold changes for qPCR data, and on the % of Fluorescent Unit / μg of Proteins for the DAN assay.

10.) A summary drawing of the regulatory loop between NO and RA would be informative, also indicating the known target genes (from this study).

We completely agree that a summary drawing is needed, so we included it in this version of the manuscript as Figure 4. We accordingly modified the text in the discussions.

[Editors' note: further revisions were suggested prior to acceptance, as described below.]

1) The authors contradict themselves in the expression of the NosB gene. They state in the present author response letter: "Nevertheless, we confirmed that NosA and NosB are not expressed at these stages…." This is a direct contradiction to Figure 2 of Annona, G et al., 2017.The reviewers and RE are aware that further research into a topic can lead to the realization that previously published data (even if acquired with all best practice and controls) are incomplete or even incorrect. However, if this happens then this needs to be transparently clarified in the following work.

We agree with the reviewer’s team that a detailed clarification is needed on this point.

In the Annona et al., (2017) we studied NosB expression using two approaches: ddPCR and WISH. We observed expression of NosB by ddPCR at gastrula 10 hpf (higher expression) and neurula 24 hpf (lower) stages. Nevertheless, we did not find any specific NosB signal by WISH at neurula 24 hpf stage, probably due to a widespread low level expression in many cells.

In our previous author response letter we have erroneously declared that NosB is not expressed in neurula at 24 hpf; in fact, we meant that we did not detect any specific NosB signal by WISH at that stage in both control and TRIM conditions.

2) Depending on what now really is the case with the expression of NosB, a further discussion by the authors on how the chosen TRIM drug treatment period (24-30 hpf) makes biological sense is required.– Under normal conditions, NosA is not expressed during the developmental period shown to be affected by an inactivation of the NOS protein by TRIM incubation. NosB is also not expressed or only very weakly. This suggests either that low levels of the NOS protein are sufficient to produce the right amount of NO at 24 hpf or that the turnover of the NOS protein is slow and still present at 24 hpf (although translated at earlier stages, where NosB expression is higher (shown for 10 hpf in Annona et al., 2017)). NO itself is probably not stable enough to stay in the tissue. The authors should discuss this point.

We are sorry for the misunderstanding on this point. NosA is not expressed during development; while NosB is expressed at gastrula stage (10 hpf, G0) and very weakly at neurula stage (24 hpf, N2) (not detectable by in situ hybridization). NosC expression starts at neurula stage (N2) until larval stage (72 hpf, L1). We realized that, regarding the NosC expression, in the result section of the Annona et al., (2017) there is a wrong sentence “NosC expression starts at pre-mouth larval stage (48 hpf). “, although in the rest of the manuscript the description of the expression pattern is correct.

With the aim to clarify this point and to avoid confusion in the interpretation of the results presented in this work, we here show the expression pattern of NosC by fluorescent in situ hybridization [24 hpf (N2), 30 hpf (N4), 36 hpf (N5), 42 hpf (T0), 48 hpf (T1), see Figure 1—figure supplement 1]. Therefore, taking this into consideration we think that the TRIM drug treatment period we choose has a biological sense because it should suppress the NO production by NosC.

We added the NosC expression pattern in (Figure 1—figure supplement 1) and clarified the expression pattern of the three genes in the first paragraph of Results.

Of course, the presence of a stable protein, produced during an earlier developmental period but still functional in 24 hpf embryos, as proposed by the reviewer, is also conceivable.

– NosA and -B expression increases after RA incubation at developmental stages, where the genes are normally not expressed or only lowly expressed. Does this mean that a possible regulation, direct or indirect, of Nos expression by RA is not occurring under normal developmental conditions? Could it indicate that this part of the regulatory loop potentially could be used at other developmental stages? – e.g. regulating NosB expression during blastula/early neurula stages or regulating NosA expression in adults (according to the expression levels published in Annona et al., 2017)? Or is this part of the regulatory loop only activated when an NO/RA unbalance occurs?

We agree that this point is crucial and need clarification. We now discuss this point in our revised manuscript and we propose that NosA and NosB expression activation by RA is not occurring under normal developmental conditions, and that, possibly, it is a mechanism protecting the anterior part of the amphioxus body against an unbalanced NO/RA ratio. However, we cannot exclude that RA treatment forces the normal gene regulation and shows up what happens in other developmental stages or in the adult. We, therefore, modified the text in the Discussion section.

NosC genomic region has many RARE's, but is not regulated by RA at the developmental time, which the authors have analyses. Maybe NosC could be regulated by RA earlier or later in development? Again, the authors should discuss this point.

Functional experiments would be necessary to answer this point. Nevertheless, following the comment “4.)”, we decided to eliminate from the text the in silico results concerning the presence of RAREs in all three Nos loci.

3) Related to the point 2 above- In answer to the discrepancy between the previous Annona paper and the current manuscript with regards to the differing developmental windows that are sensitive to TRIM treatments, the authors have explained in their response why this is, but not clearly described this in the revised manuscript. It is important to make it clear to readers that the currently reported windows are the correct ones. Without such an explicit statement that the previous Annona work is incorrect in this regard, these important details will be confusing to readers who are trying to follow the work from this lab. It has clearly confused the referees.

We agree and addressed this point in the text in the Results section “NO controls pharyngeal development during early neurulation in amphioxus“. The discrepancy between our previous Annona et al., (2017) paper and the current manuscript resides in the fact that the developmental window in the former was studied upon administration of the L-NAME, that is an L-Arginine analog, while in the present work we preferred to study the developmental window using the Nos enzyme inhibitor TRIM. Thus, the dissimilarity in the time window of active inhibition between the two treatments can be explained by the significant difference in Nos inhibition efficiency between the two molecules. Indeed, TRIM is a more potent Nos inhibitor and is effective at much lower concentrations than L-NAME, which probably justifies why a shorter treatment duration leads to the pharyngeal defect phenotype when we use TRIM, and hence the apparent discrepancy with previous data.

We agree this could be misleading for the reader and we now clearly give the explanation in our revised manuscript.

4) RAREs: The authors state that "Moreover, we searched in a number of genomic fragments (50 kb long) randomly chosen and found a comparable number of putative RAREs to Nos genes loci. »This makes it very clear that there is no significant enrichment of RAREs in the Nos genes and hence the data must not be presented as support for a regulation by RA. So if the authors choose to still include this observation, it has to be presented as it is: as non-significant. For this reason, a statement such as 'RA regulates the transcription of Nos genes, probably through RA Response Elements found in their regulatory regions' (abstract) has to be removed and the according sections in the main text re-written or removed.

We agree that it is still preliminary and experimental data would be needed to support our proposition. We therefore decided to eliminate this observation from the text (results, discussion and methods). We also eliminated the “Figure 2—figure supplement 1B” and the reference Sandelin and Wasserman, 2005.

5) Figure 1 —figure supplement 1A (+ lines 90-91):– Here, the effect of TRIM is evaluated by measuring the pharyngeal length. Optimally, the total body length should also be taken into account, so the comparison would be of the pharyngeal length/the total body length. What if the larvae are overall shorter, then maybe there is no difference in the pharyngeal length with respect to the total body length in the TRIM treated larvae (but yes in the morphology and position of the pharyngeal structures).

Following the reviewer’s comment we measured the pharynx length in comparison to the total body length, and it results that, although the total body length was unaffected after treatment, the pharynx was actually shorter in TRIM-treated larvae. Specifically, the ratio between the pharyngeal length/the total body length in TRIM condition came out to be reduced by 32% in average compared to controls (min 14% – max 50%). We have now included the pharynx/body ratio in Figure 1 —figure supplement 2A, and modified the text and the corresponding figure legend accordingly.

6) In Figure 1-Supplement 1B: Clearly indicate at which time point the CTRL was taken. Furthermore, the most appropriate way to perform this experiment would be to take a control sample at the beginning and the end of each the different time windows of treatment. Ideally you could add this. If not, please carefully explain why this was not done and why the data are nevertheless meaningful.

We now clearly show that the control was taken for each time point (30 hpf, 36 hpf and 42 hpf) and we modified the Figure 1—figure supplement 2B and its legend accordingly.

The same batch of embryos at 24 hpf was divided in two: the control half was incubated with DMSO while the other half in TRIM. In this way, the NO level at the starting point of the experiment in the two samples was identical.

We agree that this experiment could have been done in the way reported by the reviewers, being more informative about the trend of NO reduction; nevertheless, by recording the NO level at specific (end) time points, after 6, 12, and 18 hours of TRIM incubation in controls and treated embryos, we have been able to demonstrate the efficiency of the treatment.

We modified the manuscript in the first paragraph of the Results.

7) Figure 1B: is the scale of these reconstructions the same between panels? It looks as though it is not, and that the TRIM-treated images might have been slightly enlarged relative to the controls. If this is the case, the authors need to change this so that everything is at the same scale. As currently presented, it is difficult for readers to appreciate the shortening of the pharynx region in TRIM-treated larvae (it certainly does not look like 40% shortening in Figure 1B).

The picture scale is identical in all panels of the Figure 1B. As said in our answer to point ”5.)” the reduction of the ratio pharynx length/body length in TRIM-treated larvae versus controls is of 32% on average, with a minimum value measured of 14% and a maximum value of 50%. The larva showed in Figure 1 lay within the frame of the measured samples.

8) Line 105-106: The authors claim that the 'critical time of its [NO] action is restricted approximately to the first six hours (24-30hpf)'. Given that incubations starting at 36h still result in 75% defective embryos, this claim is factually wrong.

We agree and improved the sentence accordingly. The NO action time on pharynx development is indeed comprised between 24 (N2) and 42 hpf (T0), but we meant to say that 24-30 hour (N2-N4) is the minimal time window of NO signaling disruption that we have identified causing pharynx malformations.

Therefore, we modified the text in the first paragraph of Results.

9) Figure 1-supplement 4: is it correct that the authors compared Cdx, Six1/2 and IrxC in the RNAseq and RT-PCR panel, but then Cdx, Six1/2, Pitx in the ISH panel, or should this be the same three genes in both panels?

To be more consistent we decided to show the same genes in the RNA-seq and qRT-PCR panel and in the WISH panel. So we included Pitx, Six1/2, IrxC, and Cdx in Figure1—figure supplement 5 (C-C’ and D). Accordingly, we added in Supplementary File 1 the primers of Pitx for the qRT-PCR and of IrxC for the preparation of the RNA probe for WISH experiments.

10) With the new inclusion of genes that are supposedly not regulated by NO/RA in the stages of development examined in this revised manuscript, something striking has become apparent. Is NO inhibition with TRIM supposed to be posteriorizing the embryos as a whole, or in a more restricted fashion (either in space or time)? If it were a general posteriorization, why then do Cdx, Pitx and Six1/2 not shift? Particularly since amphioxus Cdx is a gene that has previously been shown to be sensitive to changes in RA levels (Osborne et al., 2009, Developmental Biology 327: 252-262). Is this simply due to not enough time elapsing between the TRIM treatments performed here, the response of the first suite of genes involved in RA signaling, but then not the next tranche of responsive genes downstream of RA? This, however, does not seem to be the entire (or even likely) explanation, as the authors RA manipulations supposedly phenocopy the TRIM treatments.Instead, do the different effects of manipulating RA levels in the current study relative to previous studies reflect the different developmental times that RA levels are altered between studies, thus producing shifts in A-P regionalised genes to different extents or in different regions between different studies? Perhaps this relates to the temporal shifting of the RA production centre as amphioxus development proceeds, this shifting process already being briefly mentioned by the authors, and this could act in conjunction with the NO signaling centre not shifting but remaining in the pharyngeal region (if this is the case)?

We agree that it might appear that there are discrepancies between the effect of treatments between our study and others. However, in the present study we are not analyzing the effect of RA treatment at the phenotypical level. We only used RA treatment to test for its effect on expression of genes for which expression is also affected by TRIM. What we show is that TRIM treatment partially phenocopies the RA treatment with similar effects at the level of pharyngeal development and at the level of RA pathway genes expression. However, TRIM treatment indeed does not lead to a similar effect as RA treatment concerning the whole body posteriorization.

In this context, the comment on line 225 about "posteriorization of the larval body plan" is misleading if the shift is not actually in the entire body plan but only in the pharyngeal region.

At this point of the manuscript we were talking about the shift of the anterior limit of Hox1 and Hox3 known to indicate the posteriorization of the body plan.

However, as said before, we agree that the observed phenotype is a posteriorization of the pharyngeal region and we have modified the text in the appropriate places.

These observations warrant a brief discussion of A-P patterning in amphioxus to clarify more precisely how the authors think the phenotypes they observe are generated in the context of amphioxus development more widely, and why their phenotype seems to be restricted to the pharynx region when RA-linked phenotypes are much more widespread in amphioxus.

We improved the text regarding the A-P patterning in amphioxus and effects of RA and TRIM treatments.

Further suggestions for improvement:Figure 4B:A modification of the summary figure (Figure 4) would facilitate the reading of this figure. Now a decrease or an increase of a molecule (NO, RA) or a gene product is indicated in many ways: thickness of the arrows between two components, the size of the molecule or gene product and the addition of red or green arrows to show a decrease or an increase, respectively. The authors need to make this figure clearer e.g. through 1) showing all connecting arrows in one size and 2) use only one type of arrows, so no arrows with dashed lines (or explain better in the figure legend). A distinction in arrows could be based on the process (enzymatic process (Nos -> NO and Aldh1a -> RA) vs transcriptional regulation (NO -> Aldh1a, RA -> Nos and RA -> genes of the RA machinery)). 3) Choose either to show a decrease or an increase by the size of the molecules and gene products OR by adding the green (increase) and red (decrease) arrows, but not both. It's to much information. 4) Remove the thick arrow between Figure 4B and 4C.– The authors need to indicate what causes the decrease in NO levels in Figure 4B e.g. by placing a cross over the arrow between Nos and NO to illustrate an inactivation of Nos.– add a question marker to the arrow between NO and Aldh1a to indicate that it is not known exactly how this regulation takes place.– instead of "RA levels reduction" maybe write "retinoid metabolism". That RA levels are reduced is indicated with the arrow.– For consistency, the authors could write "Normal state" over Figure 4C as for Figure 4A, or "Recovery of normal state".

We agree that the summary scheme of Figure 4 needed a simplification. Therefore, we adopted all the suggestions indicated and modified the figure legend accordingly.

line 87:– regarding the treatment window from N2 to T2. Shouldn't this be N2 to T3?

The treatment was performed up to 48 hpf (T1). We followed the recently published staging system by Carvalho et al., (2021) and therefore considered the following correspondences of hpf and developmental stages used in our experiments (18°C): 24 hpf = N2, 30 hpf = N4, 36 hpf = N5, 42 hpf = T0, 48 hpf = T1, 72 hpf = L1.

line 90-94:– revise here all references to the figure e.g. in line 92 and 93, the reference should be Figure 1B panel I-II and V-VI (and not Figure 1B, panel I-II and III-IV). References to part of the figure 1B are missing in the text (Figure 1B III, IV, VIII). The authors should include these e.g. referring to the other matching panels, thus refer to panel I-III, II-IV, V-VII and V-VIII together, respectively.

Sorry for this, we corrected the figure citations appropriately.

line 99:– add figure reference to Figure 1A after "…70% affected larvae" to facilitate reading.

Done.

line 101:The authors write that 'NO levels ….dropped to 40% in comparison to controls'. As the '40%' is based on comparing average concentrations, the authors should report a range rather than an exact number.

Right. We give more information in the revised version saying that NO concentration decreased to 40% in average (in a range between 58,5% and 18%).

line 135:(Figure 2A, D) and not "(Figure 2A, B)".

Done.

line 194:correct to "described"

Done.

line 322-324:– Add "shown indirectly" to be clear: "A reduction of NO (Figure 4B), as shown by our results, induces an excess of RA, shown indirectly through the overexpression of Aldh1a."

Done.

line 346 Key Resources Table:– remove the column "additional information" if there are no additional information.

Done.

line 660: write Cyp26.2 in italic

Done.

Figure 1 —figure supplement 1A.– length not "lenght" on the y-axis

Done.

There is still some further editing and proofing of the written English required. There are a handful of odd word choices (e.g. change sensible/insensible to sensitive/insensitive) and slightly awkward sentence constructions.

A native English speaker has carefully edited the manuscript.

[Editors' note: further revisions were suggested prior to acceptance, as described below.]

Recommendations for the authors on the revised manuscript:The manuscript reads very well now, but I think some of their main suggestions and discussions have to be revised to support the main message of the manuscript. Here I'm referring to the following observations and statements by the authors:1. NosC is responsible for the production of NO from N2 to L1 larval stage (line 101-103).2. RA and TRIM treatment experiments cause an increase in NosA and NosB expression, but not in NosC (line 194-205, Figure 2E) – why they conclude:"A possible explanation could be that while in normal conditions RA does not regulate NosA and NosB expression, it directly or indirectly regulates such gene expression in the case of a NO/RA inbalance as a way to restore the correct NO/RA ratio."(line 343-345)Based on these conclusions, the title of the manuscript "Crosstalk between Nitric Oxide and Retinoic Acid pathways is essential for amphioxus pharynx development" is not physiological relevant. According to the authors, the crosstalk only occurs when an NO/RA unbalance takes place. To me, this is a problem.

We do not agree with the reviewer on this point. We think that the NO/RA crosstalk occurs in vivo during development, and indeed the NO/RA imbalance falls within a possible physiological scenario taking into account that during development natural mutations in the RA machinery (synthesis or catabolism) could alter the correct endogenous RA content or gradient. Therefore the default state is that NO controls RA pathway, while the RA regulation of NO synthases plays as a natural mechanism of compensation in case of NO/RA imbalance. This kind of mechanism, usually only revealed by artificial signaling pathway modifications, provides robustness to developmental processes, and is hence essential for correct embryogenesis.

I think the authors are missing to include another plausible possibility, which would give biological meaning to the work and which they actually do discuss in the Response to Review Comments: NosB could also contribute to NO production during normal pharynx development (this could still be together with NosC). The results from Annona et al. 2017 supports this: Expression levels validated by ddPCR of NosA, NosB and NosC are shown in Figure 2h-h'. When expressed, both NosA and NosB show ~20x higher expression levels than NosC – also when comparing expression levels between NosB and NosC at 24 hpf, which is when they initiated the 6 hours TRIM treatment. Now of course, it would be interesting to have the in situ hybridisation for NosB at 24 hpf. In Annona et al., 2017 they show the expression pattern until mid-gastrula stage and not during the Nerula period (TRIM treatment starts at N2). In the Response to Review Comments, the authors declare that they did the in situ hybridisation for NosB at 24 hpf and didn't detect any pattern. They argue that this could be because of a broader weaker expression. This makes sense. I think they should mention this in the manuscript. And also that the NosB expression during mid-gastrula was a broad expression in the ectoderm, next to the endoderm (where I guess the pharyngeal tissues develop from). Including the possibility that NosB could contribute to NO production during pharynx development would give a biological meaning to the conclusion that there is a crosstalk between NO and RA pathways during amphioxus pharynx development. And not that the crosstalk only takes place during a NO/RA inbalance relying on the activity of NosA and NosB, which are normally not expressed during pharynx development (as stated in the current version of the manuscript, but only partially true as mentioned above).

We included in the Results paragraph the concept of the possible contribution by NosB to the NO production in the selected time window. Nevertheless, we decided not to go deeper into NosB expression pattern discussion because it goes beyond the scope of this work. NO signal may be necessary during gastrulation for the specification of the pharynx; nevertheless, in absence of dedicated experimental proofs, it is highly speculative at this stage. Therefore we decided not to add any discussion about this point.

In this regard, maybe a short discussion on NO production site (based on NosB and NosC expression) vs NO target tissues (Aldh1a expression) could help clarify the above concern.

We think this is a very interesting point as well although to address this issue it is necessary to perform ad hoc experiments, as for example to perform a double fluorescent in situ hybridization of NosC and Aldh1a. For this reason we will certainly deepen this topic in our next work.